# Machine learning and structural analysis of *Mycobacterium tuberculosis* pan-genome identifies genetic signatures of antibiotic resistance

Erol S. Kavvas [1], Edward Catoiu[1], Nathan Mih[1,2], James T. Yurkovich [1,2], Yara Seif [1], Nicholas Dillon[3,4], David Heckmann[1], Amitesh Anand[1], Laurence Yang[1], Victor Nizet [3,4], Jonathan M. Monk[1] & Bernhard O. Palsson [1,2,3]

*Mycobacterium tuberculosis* is a serious human pathogen threat exhibiting complex evolution of antimicrobial resistance (AMR). Accordingly, the many publicly available datasets describing its AMR characteristics demand disparate data-type analyses. Here, we develop a reference strain-agnostic computational platform that uses machine learning approaches, complemented by both genetic interaction analysis and 3D structural mutation-mapping, to identify signatures of AMR evolution to 13 antibiotics. This platform is applied to 1595 sequenced strains to yield four key results. First, a pan-genome analysis shows that *M. tuberculosis* is highly conserved with sequenced variation concentrated in PE/PPE/PGRS genes. Second, the platform corroborates 33 genes known to confer resistance and identifies 24 new genetic signatures of AMR. Third, 97 epistatic interactions across 10 resistance classes are revealed. Fourth, detailed structural analysis of these genes yields mechanistic bases for their selection. The platform can be used to study other human pathogens.

[1] Department of Bioengineering, University of California, San Diego, La Jolla, CA, USA. [2] Bioinformatics and Systems Biology Program, University of California, San Diego, La Jolla, CA, USA. [3] Department of Pediatrics, University of California, San Diego, La Jolla, CA, USA. [4] Skaggs School of Pharmacy and Pharmaceutical Sciences, University of California, San Diego, La Jolla, CA, USA. Correspondence and requests for materials should be addressed to J.M.M. (email: jmonk@ucsd.edu) or to B.O.P. (email: palsson@ucsd.edu)

Advancements in genome sequencing technologies have made available thousands of drug-tested *M. tuberculosis* genomes in public databases. With available sequences expected to surpass 60,000 during the next 5 years (https://www.crypticproject.org/), there is impetus for new quantitative approaches that excel at analyzing massive datasets. Methods that explicitly account for structure amongst features—such as those found in the field of machine learning—will be essential for addressing this *M. tuberculosis* data deluge[1].

To date, most approaches compare *M. tuberculosis* genome sequences against the H37Rv reference strain in order to identify single nucleotide polymorphisms (SNPs). Following SNP identification, most studies then focus on the subset of previously identified resistance-determining SNPs that have been previously determined to be key resistance-determining mutations, specifically those within a handful of genes encoding proteins targeted by drugs[2]. While such studies have proven to be powerful for diagnostics[3] and elucidating mutational steps to AMR[2], they do not account for potential genome-wide mutations reflecting positive AMR selection, such as those found to be related to cell wall permeability, efflux pumps, and compensatory mechanisms[4].

Specific genome-wide functional analyses in *M. tuberculosis* have shown that *ald* loss-of-function[5], *ubiA* gain-of-function[6], and *thyA* loss-of-function[7] mutations occur in off-target reactions, and confer resistance through modulation of metabolite pools. These results exemplify the complex interplay underlying AMR phenotypes that extends beyond the few genes currently utilized in diagnostic studies. In addition to limitations of a narrow genetic view, the identification of other types of resistance-conferring mutations, such as deletions[8,9], suggest that SNPs are no longer comprehensive in describing the mutational landscape of *M. tuberculosis* AMR evolution.

Here, we apply a reference-agnostic machine learning approach complemented by both genetic interaction and protein structural analysis to deduce the variability in genetic content and AMR of 1595 *M. tuberculosis* strains. The complete analysis recapitulates known AMR mechanisms and infers specific selection pressures through directed hypotheses.

## Results

**Characterizing the *M. tuberculosis* pan-genome.** Our first goal was to characterize and understand the gene content of sequenced *M. tuberculosis* strains. We selected a representative set of 1595 *M. tuberculosis* strains for which AMR testing data was available from the PATRIC database[10] and come from a wide range of studies (see Supplementary Discussion). Strains were selected for their genetic, geographic, and AMR phenotypic diversity (Supplementary Fig. 1). The geographic diversity of these strains reflects areas heavily burdened by *M. tuberculosis* (Supplementary Fig. 1a). We constructed a phylogenetic tree for the 1595 strains using a robust set of lineage-defining SNPs[11] (Supplementary Fig. 1b and Methods). Finally, strains were selected in order to provide a distribution across commonly used *M. tuberculosis* treatment regimens (Methods). Of these 1595 strains, 1282 strains had AMR testing data for isoniazid, rifampicin, streptomycin, and ethambutol (Supplementary Fig. 1c) and 946 (59%) were resistant to both isoniazid and rifampicin. Following the selection of strains, we determined the pan-genome (i.e., the union of all genes across the strains) represented by these data and analyzed the distribution of various genomic features (core genes, virulence factors, etc.). The pan-genome analysis described a general theme of high conservation (Supplementary Fig. 2, see Supplementary Discussion for further discussion of *M. tuberculosis* pan-genome).

**Assessing allele frequencies identifies key AMR genes.** Although the *M. tuberculosis* pan-genome clusters provide an informative view of the global genetic repertoire within a species, they lack the resolution necessary to discriminate between most AMR phenotypes. To elucidate fine-grained genetic variation indicative of AMR evolution, we separated each pan-genome cluster into groups of exact amino acid sequence variants, or alleles (Supplementary Fig. 3g). In contrast to alignment-based perspectives, the allele-based pan-genome does not reduce non-*H37Rv* variants to a collection of SNPs, but instead represents variants in their functional protein-coding form. This approach accounts for all protein-coding alleles in the *M. tuberculosis* pan-genome, thereby representing the extensive strain-to-strain variation observed in bacterial genomes without biasing the variations relative to a single reference genome.

We used mutual information (MI)[12] as an association metric to identify resistance-determining genes with this newly constructed variant pan-genome and the accompanying AMR dataset (Methods). Importantly, this approach identified primary resistance-conferring genes previously reported in the literature (Fig. 1). In addition to MI, we calculated associations using a chi-squared test and an ANOVA *F*-test, both of which identified similar sets of key AMR genes ($P < 0.005$; Bonferroni correction) (Supplementary Data 1). These results suggest that allele frequencies based on exact sequence (i.e., without a metric for genetic distance) are capable of identifying AMR genes, which has previously been shown with k-mer based approaches[13–15].

**Machine learning identifies known and new resistance genes.** Although simple and effective, pairwise association tests (i.e., MI, chi-squared, and ANOVA *F*-test) do not simultaneously account for multiple alleles because the pairwise calculations consider variants independently of one another. Thus, we tailored a support vector machine (SVM)—a method that inherently accounts for structure amongst the features—to uncover AMR-conferring genes (Methods). Using the allele presence–absence across strains as the features, the SVM identified an additional seven known AMR gene–antibiotic relations absent from the top 40 ranked alleles determined by pairwise associations, including those associated with complex resistance (Table 1). In particular, *ubiA*, a resistance gene recently found to confer high level resistance to ethambutol[6], appeared as a strong signal across the ensemble of SVM simulations—despite not being accounted for in contemporary *M. tuberculosis* diagnostics (Supplementary Data 2).

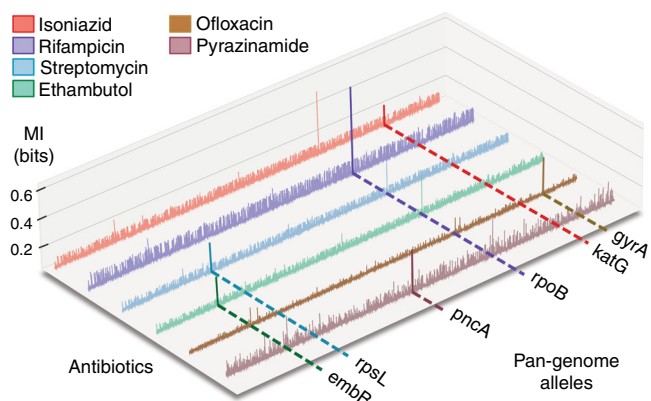

**Fig. 1** Identification of key resistance-conferring genes using mutual information. The pairwise mutual information (vertical axis) between the pan-genome alleles and antibiotic resistance was calculated across all possible pairs. The listed genes correspond to the pan-genome alleles that hold the most information about the listed drug's AMR phenotype

**Table 1 Known AMR genes uncovered by machine learning**

| Antibiotics | Known AMR genes |
|---|---|
| Isoniazid | katG[43], inhA[a,20], fabG1[44] |
| Rifampicin | rpoB[45], rpoC[a,46] Rv3239c[47] |
| Ethambutol | embB[48], embC[17], ubiA[a,6], embR[a,17] |
| Pyrazinamide | pncA[49] |
| Streptomycin | rpsL[50], gidB[51] |
| Ofloxacin | gyrA[52] |
| 4-Aminosalicylic acid | folC[a,7], thyA[a,53] |
| Ethionamide | ethA[54], inhA[a,20] |
| Known AMR genes associated with other antibiotics | dprE1[55], ald[5], alr[56], murA[57], pks2[58], pks12[59], ppsA[60], ppsD[60], drrB[61], drrC[61], moeW[55], Rv0687[62], mshD[63], gyrB[52], Rv1877[64], Rv0194[65] |

The eight antibiotics shown each have an AUC greater than 0.80 (Supplementary Fig. 5)
[a]Not found in top 40 ranked alleles determined by mutual information, chi-squared, and ANOVA $F$-test

The SVM method revealed an abundance of AMR-implicated genes involved in metabolic pathways (119/317, 37.5%) (Supplementary Data 2). In fact, the majority of the known AMR determinants are metabolic enzymes (24/33, 73%). We found over 20 genes related to cell wall processes (26/317, 8.2%), which is consistent with previous findings of convergent AMR evolution in *M. tuberculosis*[4]. Furthermore, many high-signal AMR genes, such as *pbpA* and *mmpS3*, have recently been identified as determinants of intrinsic *M. tuberculosis* AMR[16]. The full list of identified genes for each drug is provided (Supplementary Data 2).

**Machine learning uncovers genetic interactions.** Beyond identifying AMR genes, four key attributes of our ensemble SVM learning approach enable analysis of genetic interactions underlying variable AMR phenotypes (Methods and Supplementary Fig. 4): (1) the weighting of a particular allele in a specific SVM hyperplane scales with its contribution to a particular AMR phenotype, (2) the sign of the weighting (positive or negative) corresponds to the contribution of that allele to the AMR phenotype (i.e., positive weights correspond to resistance while the negative weights correspond to susceptibility), (3) the magnitude and sign of an allele weighting is dependent upon the magnitudes and signs of other alleles within the same hyperplane, and (4) the use of bootstrapping (i.e., randomized subsampling of the population with replacement), and stochastic gradient descent ensures variability in the weights, signs, and set of alleles for each SVM hyperplane. Motivated by attributes 3 and 4, we hypothesized that two genes may interact if the weights, signs, and appearance of their alleles are significantly correlated across the ensemble of SVM hyperplanes (Methods). Therefore, to identify genetic interactions contributing to AMR in *M. tuberculosis* strains, we constructed a correlation matrix of allele weights across the ensemble of randomized SVM hyperplanes (Supplementary Data 3) and filtered for the top 60 highest gene–gene correlations for eight AMR classifications. The resulting set of gene–gene pairs were interrogated through logistic regression modeling, selecting those gene pairs with statistically significant allele–allele interactions ($P < 0.05$; Benjamini–Hochberg correction) (Methods and Supplementary Fig. 4). This approach uncovered 94 potential genetic interactions (Supplementary Fig. 4).

We can use the evolution of ethambutol resistance as a case study to examine the output of our approach. Epistasis analysis of ethambutol AMR genes implicated interactions between *embB*, *ubiA*, and *embR*; all genes known to contribute to ethambutol

resistance[6,17,18]. Although the *embR* alleles appeared few times across the multiple SVM simulations, their appearance was highly correlated with alterations in the sign and weight of the *ubiA* allele (see Supplementary Figure 6). This implies that *embR* is only a predictive feature within the context of *ubiA*, which may result from the weak penetrance of *embR* alleles within *M. tuberculosis* (Fig. 2a). Logistic regression modeling identified significant allele–allele interactions between *ubiA* and *embR* alleles (Supplementary Fig. 4). We investigated these interactions through a co-occurrence table of the genes, where each cell corresponds to the number of resistant strains with both alleles over the total number of strains with both alleles (Fig. 2a). The log odds ratio (LOR)—a measurement of the association of the co-occurrence of both alleles with AMR phenotype—was used to color each cell in the co-occurrence table (Fig. 2, see Methods). We observed that the resistant-dominant *ubiA* alleles (i.e., those with high positive LOR), 2 and 4, occurred exclusively in the background of nonsusceptible-dominant *embR* alleles (Fig. 2a). Interestingly, in contrast to *embB* and *ubiA*, no *embR* allele appeared as a clear resistance determinant (Fig. 2a). Furthermore, neither *embR* nor *ubiA* were significantly associated with ethambutol AMR in pairwise associations tests (Table 1 and Supplementary Data 1), showing that our ensemble-based machine learning approach uncovers *M. tuberculosis* AMR complexity. In addition to these known AMR determinants of ethambutol, our analysis implicated *ubiA* interactions with *Rv3848* in ethambutol resistance (Table 2 and Supplementary Data 4). Interestingly, the resistant-dominant allele of *Rv3848* occurs exclusively in the background of the AMR-neutral *ubiA* allele 3, hinting at an alternative route of high-level ethambutol resistance.

For identified isoniazid AMR genes, the co-occurrence table highlighted cases where either *katG* or *inhA* genes provide the dominant mode of resistance (Fig. 2b). Specifically, the incidence of susceptible *katG* alleles 1, 2, 5, and 6 (i.e., low LOR) with the resistance *inhA* alleles 2 and 3 (i.e., high LOR) showed that isoniazid resistance in our dataset arose from either *katG* or *inhA* mutations, but not both. Aside from these two highly studied isoniazid AMR determinants, epistatic interactions between *katG* and *oxcA* appeared with a high signal and further displayed an interesting co-occurrence relationship with *katG* (Fig. 2b). This epistatic interaction for *oxcA* has not been previously described; specifically, alleles 3 and 7 of *oxcA* appear exclusively in isoniazid-resistant strains. While the AMR phenotypes for the strains containing these alleles may be attributed to the presence of the resistance-dominant *katG* alleles 3 and 7, as is often offered in studies to "explain resistance", the variation in AMR phenotypes across the different alleles were determined to be significant by the machine learning algorithm and thus motivated further investigation. Co-occurrence tables of epistatic AMR genes are provided for the ten antibiotic classifications (Supplementary Data 5).

**Structural analysis suggest drivers of selection.** Although the machine learning results agree with experimental literature, it remains unclear whether the uncovered genetic features are either true determinants of AMR or possible artifacts of the statistical learning algorithm. To gain additional insight into whether or not the uncovered alleles are causal in AMR evolution, we mapped the alleles of the 254 AMR genes to protein structures using both experimental crystal structures (20/254) and predicted homology models (50/254) using the ssbio Python package (Methods and Supplementary Data 6)[19]. Out of the 254 genes, 217 had available protein sequence annotations (i.e., binding domains, secondary structures, etc.). First, we established a positive control by

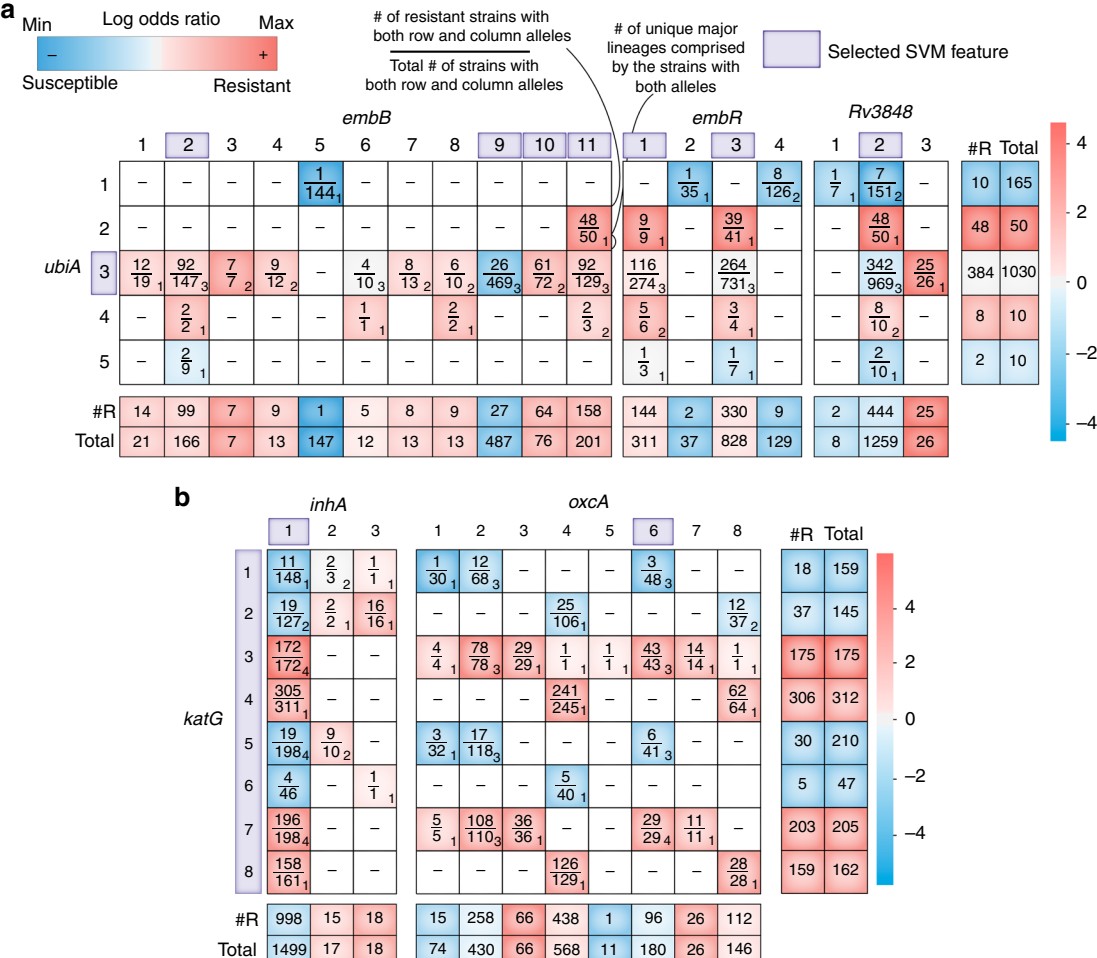

**Fig. 2** Allele co-occurrence tables of correlated AMR genes. Co-occurrence of epistatic genes identified in **a** ethambutol and **b** isoniazid. For the rows on the bottom and on the far right, #R refers to the total number of strains that have the allele and are resistant to the specific drug. Total refers to the total number of strains that have that allele that were tested on that specific drug. Each cell is colored by the log odds ratio (LOR) with respect to the AMR phenotype. The numbers in the bottom right of each allele co-occurrence box describes the number of unique sublineages comprised by the strains with both alleles (Methods). The alleles enclosed by a purple box represent those chosen as features by the support vector machine (SVM). Note that in some cases the rows and columns do not sum up to the total strains due to rare cases when strains lack those alleles (Methods)

mapping the alleles of known AMR genes to protein structures and verified that resistance-conferring alleles were located in annotated structural regions that indicate the known mechanism of action (Supplementary Fig. 7). For example, structural mapping of the isoniazid AMR-determinant, *inhA*, showed that the resistance-dominant alleles of 2 and 3 are located within two NAD-binding domains (Fig. 3a). The incidence of these two alleles in proximal NAD-binding domains is congruent with the experimentally derived mechanism of action, which describes the bactericidal effect of tight binding between the isoniazid-NAD adduct and *inhA*[20,21]. Moreover, the resistance-conferring mutations in the NAD-binding domains explains the previously described allele co-occurrence of susceptible *katG* alleles 1, 2, 5, and 6, with resistant *inhA* alleles 2 and 3, because the isoniazid-NAD adduct results from binding to *katG*, which would only occur if the *M. tuberculosis* strain lacks the resistance-conferring *katG* mutation that disables the isoniazid binding opportunity. With established confidence through case–controls, we set out to analyze the implicated and uncovered AMR genes.

Revisiting the ethambutol case study, we noticed that the susceptible-dominant *embR* alleles shared an SNP that is 14.6 Å away from the DNA-binding domain (Fig. 3a). Given that *embR* is a positive regulator of *embB*[22] and that the expression of *embB*

decreases in the presence of ethambutol[6], the SNP suggests a relative increase over alleles 1 and 3 in expression of the ethambutol target, *embB*, through increased DNA binding. For *oxcA*, the resistance-dominant alleles, 3 and 7, uniquely share mutations at residue 253, which is contained in the thiamin diphosphate-dependent enzyme M-terminal domain and is 4.51 Å proximal to a mutation at residue 224 shared by most alleles (Fig. 3). Notably, *oxcA* is an essential oxalyl-CoA decarboxylase enzyme that converts toxic oxalyl-CoA to $CO_2$ and formyl-CoA, and plays a role in low pH adaptation in *E. coli*[23]. The totality of studies describing the poisonous effect of glyoxylate[24], significant acid stress in the macrophage environment, use of $CO_2$ as a carbon source[25], and the importance of glyoxylate metabolism in antibiotic tolerance[26], all suggest that the uncovered resistance-conferring adaptations in *oxcA* increase depletion of oxalyl-CoA through increased binding affinity of the thiamin diphosphate cofactor. Without structural models, sequence annotations of structural features enabled the delineation of resistant and susceptible allele mutations to unique structural domains—highlighting an advantage of our exact-variant perspective (Fig. 3b). We provide a list of newly implicated AMR genes along with their associated antibiotic, key mutation frequency, and structural protein features (Table 2).

**Table 2 Newly proposed AMR genes**

| Gene | Drug | Dominant allele | Mutation | Structural domain feature |
|------|------|-----------------|----------|---------------------------|
| *Rv3848* | EMB, XDR | R: (25/26) | SNP | Outside transmembrane helical domain |
| *embR* | EMB | S: (2/37, 9/129) | SNP | Proximal to DNA-binding domain |
| *Rv3129* | EMB | R: (8/11) | SNP | – |
| *proC* | EMB | S: (1/27, 11/127) | SNP | – |
| *kdpC* | EMB | R: (80/91) | SNP 11 | Inside transmembrane helical domain |
| *oxcA* | INH | R: (66/66, 26/26) | SNP 253 | TPP enzyme M-terminal domain |
| *chp2* | ETA | R: (29/37, 34/60) | SNP 296 | DELs in mutagen and helical domain |
| *lipD* | ETA | R: (48/58, 8/12) | SNP 105 | Inside beta-lactamase domain |
| *Rv3471c* | ETA, XDR, SM | R: (48/50) | SNP 64 | Inside Cupin 1 domain |
| *mmpL11* | PAS | R: (35/48) | SNP 520 | – |
| *Rv0044c* | PAS | R: (13/13) | DEL 137–264 | BAC Luciferase |
| *Rv0954* | PAS | R: (34/46, 4/6) | SNP 223 | Different mutational backgrounds |
| *Rv2560* | PZA | S: (6/41) | DEL 1–80 | Compositional bias Proline-rich domain |
| *Rv2090* | RIF, INH | S: (9/67, 6/46, 5/51) | SNP 295 | – |
| *lpqZ* | RIF | S: (10/91, 12/79) | SNP 119 | Within opuAC signaling domain |
| *Rv1597* | RIF, MDR, INH | R: (18/19) | SNP 196 | No mutation in methyltransferase domain |
| *Rv1543* | RIF, MDR | S: (10/84, 12/80) | SNP 128 | Proximal to binding domain |
| *nuoL* | MDR, PAS | R: (17/17) | SNP 503 | Outside transmembrane helical domain |
| *dnaA* | SM | R: (22/22) | SNP 233 | Proximal to nucleotide binding domain 213 |
| *yajC* | SM | R: (30/30) | SNP 87 | Within transmembrane helical domain |
| *accD5* | OFX, MDR | R: (16/16) | SNP 127 | Within CoA carboxyltransferase domain |
| *Rv3041c* | RIF, OFX, SM, MDR | R: (20/28, 25/44) | SNP 140 | SNP in ATP binding domain |
| *VapC21* | XDR | R: (14/23, 14/20) | DEL 88–138 | Within second magnesium binding domain |

The mutation column represents the distinguishing mutation for the resistant or susceptible-dominant allele(s). Abbreviations: R, resistant; S, susceptible; EMB, ethambutol; PAS, para-aminosalicylic acid; INH, isoniazid; PZA, pyrazinamide; RMP, rifampicin; SM, streptomycin; OFX, ofloxacin; ETA, ethionamide; MDR, multidrug resistant; XDR, extensively-drug resistant

**Resistant and susceptible alleles are globally stratified**. Since our set of *M. tuberculosis* strains spans multiple continents, we geographically contextualized our set of SVM-derived AMR genes towards delineating possible country-specific adaptations (Table 2). We observed that resistant and susceptible alleles of the identified AMR genes were stratified amongst specific countries of origin: resistant-dominant alleles were primarily located in Belarus, South Africa, and South Korea, while susceptible alleles were primarily located in India (Table 2). The geographic locality of ethambutol, rifampicin, and isoniazid resistant alleles suggests a genetic basis underlying the successful proliferation of *M. tuberculosis* in Belarus—a country with the highest prevalence of multidrug resistant (MDR) strains ever recorded[27]. We observed that the resistant alleles associated with para-aminosalicylic acid (PAS) were based in the high-burden MDR country of South Korea. Since PAS was a key component in the standard MDR treatment regimen of South Korea[28], these alleles may represent specific adaptations to post-MDR PAS treatment that could be leveraged to better optimize the regimen. In total, these results portray a geographic basis for *M. tuberculosis* AMR evolution and demonstrate that our phylogenetically-agnostic machine learning approach is capable of capturing population behavior, which often confounds AMR predictions[29,30].

## Discussion

The data deluge on *M. tuberculosis* and its AMR characteristics is likely to continue unabated until all *M. tuberculosis* strains isolated from patients will be sequenced with associated metadata to guide clinical management. A reference-agnostic computational platform needs to be developed to receive, warehouse and continually analyze this data. We have taken the first step at developing a computational platform to meet this challenge. The platform was applied to 1595 sequenced strains to yield results in four categories: pan-genome properties, identification of genes conferring antibiotic resistance, their epistatic interactions, and protein structure based mechanistic insights.

The pan-genome properties derived by our computational platform reflect the current understanding of *M. tuberculosis* genetic variability. The other three categories of results are intertwined. We recovered 33 known AMR genes and uncovered an additional 24 novel genetic targets. This demonstrates the platform's ability to generate hypotheses that may expand our knowledge of the genetic basis of AMR in *M. tuberculosis*. Some of these new targets are surprising (e.g., *Rv3471c*) and some are understandable (e.g., *oxcA*), but all provide an impetus for more detailed experimental studies (Supplementary Discussion).

The third and fourth categories of results are interconnected and detail intricate features underlying *M. tuberculosis* AMR evolution. The 74 epistatic interactions revealed are new but in many cases involve known gene partners (e.g., *ubiA*). In other cases, these new epistatic interactions involve novel gene products (e.g., *Rv2090*). This novelty, reinforced by structural insights, inform a new line of experimental inquiry (Supplementary Discussion). The larger implications of these intricacies are threefold: (1) genetic background contributes to AMR phenotypic variation, but may be subtle (e.g., *embR*); (2) high-level resistance mutations are prevalent in off-target genes, such as transmembrane proteins (e.g., *Rv3848*); and (3) high-level resistance mutations localize to countries with poor *M. tuberculosis* management (i.e., Belarus). These features point to the adverse effects of prolonged treatment[31].

While our framework successfully identifies genetic AMR signatures, there are limitations to our approach that future efforts may expand upon. For one, our platform utilizes prior knowledge of known gene–antibiotic relationships and thus does not provide a means to uniquely deconvolve out an association of a region with a specific drug (Supplementary Discussion). In addition, while our structural analysis provided a foundation for hypothesizing potential evolutionary drivers, it did not provide further support to the causality of an allele. Novel statistical methods may leverage variations in structural features towards supporting causal alleles. Furthermore, our approach lacks the ability to understand systemic relationships connecting the alleles on a mechanistic level, such as interacting changes in biochemical flux.

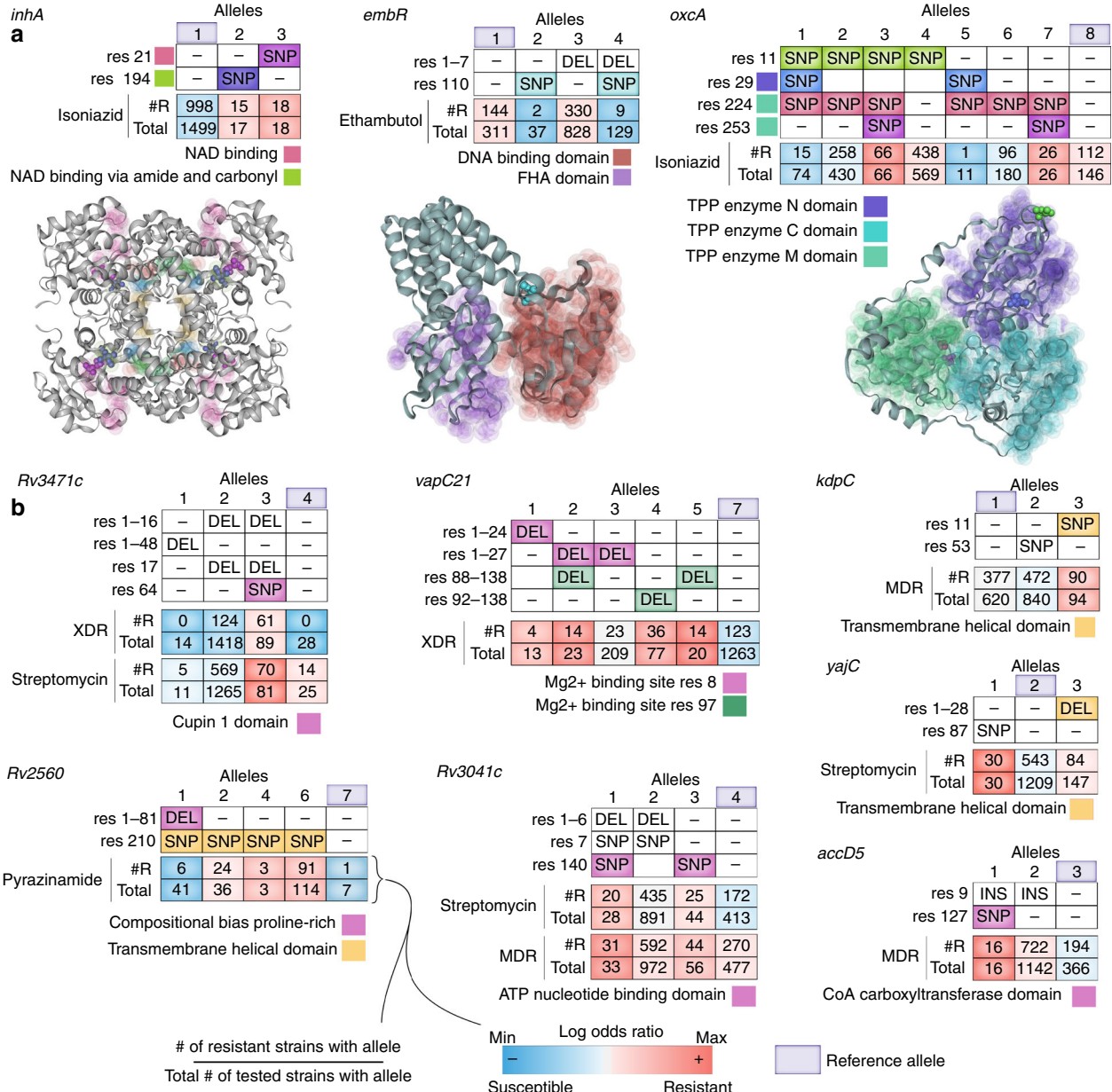

**Fig. 3** 3D and annotated protein structure mutation maps for identified AMR genes. **a** 3D protein structures with mapped mutations are shown for *inhA*, *embR*, and *oxcA*. The colors adjacent to and within the structural mutation table correspond to domains and mutations displayed on the protein structure, respectively. **b** Mutation tables for seven new AMR genes. The colors in the mutation table correspond to the incidence of an annotated structural feature located below the table. The two rows directly below the mutation table are colored according to the log odds ratio between the allele frequency and AMR phenotype. Two AMR classes are shown for *Rv3471c* and *Rv3041c*

Future efforts may integrate genome-scale models of pathogens towards elucidating and understanding the genetic signatures of antibiotic resistance[32].

Taken together, the platform presented here meets the pressing need for disparate data-type analysis enabled by rapidly growing data available for *M. tuberculosis* pathogenesis and AMR. It both recovers known AMR features (i.e., positive control) and reveals new ones. This platform utilizes a unique combination of pan-genomic analysis, machine learning, structural analysis, and geographic contextualization. These data types are likely to become available for all urgent and serious threat human prokaryotic pathogens in the near future. Similar results to those presented here are thus likely to appear on a pathogen-specific basis in the coming years.

## Methods

***M. tuberculosis* strain dataset**. The selected set of *M. tuberculosis* strains are representative of various antimicrobial resistance phenotypes, geographic isolation sites, and genetic diversity. References for the published and unpublished data sets are provided (Supplementary Discussion, Supplementary Data 7). The sequencing data for the TB Antibiotic Resistance Catalog (TB-ARC) projects (Supplementary Data 7) were generated at the Broad institute. Additional information for each of these unpublished projects can be found at the Broad Institute website. All data were acquired from the PATRIC database.

***M. tuberculosis* pan-genome construction and QA/QC**. We employed QA/QC of the constructed 1595 pan-genome by initially filtering out outlier strains. The initial selection of 1603 strains was reduced to 1595 upon review of both the cluster size distribution and the number of unique clusters across the set of all strains (Supplementary Fig. 3a, b). We found only four strains in the PATRIC database that had either a very low (<2000) or high number (>5500) of clusters. The final

selection of 1595 strains has a cluster size distribution between 3900 and 4400, and a reasonable unique cluster distribution where the number of unique clusters did not exceed 160 (note that unique is defined here as being in only one strain). The pan-genome of all 1595 strains was constructed by clustering protein sequences based on their sequence homology using the CD-hit package (v4.6). CD-hit clusters protein sequences based on their sequence identity[33]. CD-hit clustering was performed with 0.8 threshold for sequence identity and a word length of 5.

**Pan-genome core and unique cutoff determination**. We determined the core and unique pan-genome through sensitivity analysis by plotting the change in core and unique cutoff values by the change in percentage. The cutoffs were chosen to be at the point where the second derivative of the curve is the largest. The curve represents the change in pan-genome core percentage to changes in the number of strains a gene must be found in to be defined as core (Supplementary Fig. 3c, d).

**Phylogenetic tree and categorization of lineages**. We created a robust phylogenetic tree of the 1595 strains using SNPs at the core genome. Specifically, we chose a set of 2803 core genes that appeared in at least 1593 strains, included the H37Rv reference strain (83332.12). We used needle[34] to align sequences within the 2803 pan-genome clusters (a cluster is representative of a particular loci) to the H37Rv reference allele. We built a binary SNP matrix using all of the SNPs identified from the 2803 genes (21,206 SNPs in total), and then estimated a maximum-likelihood phylogeny using RaXML version 8[35]. The tree was visualized using iTOL[36].

We used an existing SNP typing scheme[11] for categorizing the strains into lineages and sublineages.

Specifically, we used a total of 141 SNPs for identifying lineages and sublineages for our 1595 TB strains. These SNPs were previously determined to be sufficient for categorizing lineages[11]. Of these SNPs, 61 were in nonsynonymous sites and the other 70 were SNPs found in drug resistance genes. These 141 SNPs comprised a total of 74 genes. The presence of SNPs were then used to categorize the strains into the defined lineages. Of the 1595 strains, 1366 strains were categorized and 229 were uncategorized. The remaining 229 strains were categorized according to their proximity to strains with lineage-defining SNPs, with proximity defined according to our core genome SNP phylogeny. We have included the frequency of lineage variants in order to help readers discern between epistatic alleles and those in tight linkage (Supplementary Data 8). Implicated co-occurring alleles that span different lineages are unlikely to be in tight linkage (i.e., hitchhikers).

For the numeric subscripts shown in Fig. 2—describing the number of unique sublineages for each allele–allele pair—were determined as the maximum number of unique sublineages at a single branch amongst all lineage/sublineage branches.. For example, an allele co-occurrence which has strains in both lineage 1.1 and 1.2 counts as two sublineages. An allele co-occurrence which has strains in both lineage 1.1, 1.1.2, 1.1.3, 1.1.3.1, 1.1.3.2, and 1.1.3.3 counts as three sublineages (1.1.3.1, 1.1.3.2, and 1.1.3.3). If an allele co-occurrence has strains in sublineages 4.1, 4.1.2, and 4.1.2.1, then only one sublineage is counted, since the strains can be traced through a single lineage (4.1 to 4.1.2 to 4.1.2.1).

**Pan-genome-wide correlation analysis**. We performed pairwise association analysis for all alleles in the pan-genome and for the 13 antibiotics to identify key AMR genes. We utilized MI, chi-squared tests, and ANOVA F-tests. MI has many statistical benefits, which include being a nonparametric method that can quantify nonlinear relationships, unlike Pearson's correlation which measures a linear relationship. MI has proven to be a natural and powerful means to equitably quantify statistical associations in large datasets[37]. The pairwise MI was calculated for each column vector in the unique variant pan-genome with each drug susceptibility vector (Supplementary Fig. 3g). The discrete entropy calculations were carried out using the Non-Parametric Entropy Estimation Toolbox (NPEET, https://github.com/gregversteeg/NPEET). Since both vectors are binary, the naive implementation of discrete entropy estimation used in NPEET is sufficient. The top 40 MI associations for 11 drugs are recorded (Supplementary Data 1).

Associations were similarly calculated with chi-squared and ANOVA tests. P values were adjusted using the Bonferroni multiple-hypothesis testing correction. Theses statistical tests and corrections were implemented using the python package, statsmodels[38]. The top 40 associations determined by chi-squared and ANOVA F-test were recorded for 10 AMR classifications (Supplementary Data 1).

**Allele feature selection through support vector machines**. The support vector machine (SVM) attempts to account for all variants together by learning a multidimensional hyperplane that best separates the susceptible and resistant strains. The resulting hyperplane is a function of all exact-variant vectors in the pan-genome. Since the goal is not to predict resistance with high accuracy, but to instead extract key insights from the data, we take a feature selection approach by gearing the linear SVM with an L1-norm penalty and stochastic gradient descent optimization algorithm using the scikit-learn package. The L1-norm enforces sparsity in the decision function, which is ideal for feature selection. The stochastic gradient descent algorithm, in conjunction with the L1-norm, returns a different set of significant features each run. Since the chosen SVM does not reach the same solution, we look at the ensemble of 200 SVM feature selection simulations.

Furthermore, we performed bootstrapping by randomly selecting a subpopulation representing 80% of the training data for each SVM simulation.

Prior to simulation, we took out the primary resistance-conferring gene of an antibiotic from the machine learning analysis of other antibiotics in order to amplify the signal of other genes—a preprocessing step previously utilized in AMR gene identification studies[5] (Supplementary Table 3). For example, all katG alleles were only accounted for as features in the machine learning analysis for isoniazid. Furthermore, we removed all mobile element proteins, PE/PPE/PE-PGRS proteins, transposases, and hypothetical proteins from consideration in the machine learning analysis due to primarily appearing in the accessory and unique pan-genome of M. tuberculosis which may confound the results. Finally, we balanced the class weight in the SVM algorithm in order to account for the imbalance of resistant and susceptible strains seen for each drug in our dataset.

Features were selected from the SVM based on a threshold value. The value was determined through tenfold cross-validation where the threshold value was optimized through grid search (Supplementary Table 3). The use of bootstrapping in the machine learning algorithm may account for biased subpopulations in the data, which often confounds GWAS analysis for M. tuberculosis[29,30].

**Filtering of gene sets for epistatic analysis**. Leveraging machine learning towards identification of genetic interactions, we constructed a correlation matrix of allele weights across the ensemble of randomized SVM hyperplanes for each antibiotic (Supplementary File 3). We limited our machine learning analysis to AMR classifications that achieved an average AUC (i.e., average area under ensemble of receiver–operator curves) greater than 0.80 (Supplementary Fig. 5). We selected the top 100 gene–gene correlations that include genes in the top 25 ranked SVM alleles for each antibiotic. We limited the correlations to in the top 25 ranked alleles in order to avoid the case when low weighted alleles appear sparsely with other low weighted alleles, which lead to significant correlations. The resulting set of gene–gene pairs were then analyzed using a logistic regression model in order to determine statistically significant interactions. The filtering of potential gene–gene pairs prior to classical quantitative epistasis analysis addresses the problem of combinatorial explosion of pairwise interaction terms in conventional techniques.

**Epistatic analysis with logistic regression models**. We utilized logistic regression to identify significant epistatic interactions. A logistic regression model was built for each potential gene–gene pair previously determined by the ensemble SVM correlation analysis. The variables of the gene–gene logistic regression model were composed of both alleles and allele–allele interaction variables:

$$Y \sim \beta_o + \sum_i \beta_i a_i + \sum_j \beta_{I+j} b_j + \sum \sum_{i,j} \beta_{I+J+k} a_i b_j, \qquad (1)$$

where $i$ and $j$ index the alleles for genes $a$ and $b$, respectively, $I$ and $J$ are the total number of alleles for genes $a$ and $b$, respectively, $Y$ is the binary AMR phenotype, $k$ indexes each unique interaction term, $a_i b_j$, and $\beta$ is the regression coefficient corresponding to each predictor. The interaction terms were limited to cases in which the two alleles co-occur in at least one strain. The interaction variable was the dot product of the two allele presence–absence vectors. In order to account for collinearity in the variables, we applied the following three filtering criteria (note that $a_i$ is interchangeable with $b_j$):

1. If the allele $a_i$ presence–absence is the same as the interaction $a_i b_j$ presence–absence, remove the $a_i b_j$ interaction variable from the logistic regression model
2. If the allele $a_i$ presence–absence is equal to allele $b_j$ presence–absence, remove both variables as well as the allele–allele interaction variable, $a_i b_j$.
3. If the allele $a_i$ presence–absence is equal to the sum of all interaction variables involving that allele (i.e., $a_i b_j$ for all $j$), remove the allele variable, but keep the interaction variables.

We filtered for allele–allele interactions with P value < 0.05 after Benjamini–Hochberg multiple-testing corrections. The resulting set of gene–gene interactions encompassing significant allele–allele interactions were portrayed through allele co-occurrence tables (Supplementary Data 5). Logistic regression and statistical tests were implemented using the python package statsmodels[38].

**Calculation of log odds ratio in allele co-occurrence tables**. The odds ratio of each cell in the allele co-occurrence tables was determined as follows:

$$OR = \frac{BR * NR}{BS * NS}, \qquad (2)$$

where BR is the number of strains that have both alleles and are resistant to the specified antibiotic, NR is the number of strains that do not have both alleles and are resistant to the specified antibiotic, BS is the of strains that have both alleles and are susceptible to the specified antibiotic, NS is the number of strains that do not have both alleles and are susceptible to the specified antibiotic. For a single allele, the odds ratio was calculated the same way with each variable representing the single allele case. If any of the four values (BR, BS, NR, and NS) were zero, 0.5 was

added to each value in order to ensure a value when computing the logarithm of the odds ratio.

**Missing alleles in allele co-occurrence tables counts**. The lack of specific alleles shown in the allele co-occurrence table is due to strains missing some alleles. For example, *embB* allele 5 is found in 147 strains but only 144 strains have both *embB* allele 5 and *ubiA* allele 2 (Fig. 2). Specifically, the three strains missing the three *ubiA* alleles are the following PATRIC strains as described by their genome identifiers: 1423432.3, 1448794.3, and 1448824.3. Searching on the PATRIC database for either *ubiA* or *Rv3806c* results in 0 hits for these organisms. While it is unlikely that the strain is missing this allele, these limitations are not due to the analysis but instead results from the selection of strains. These events happen quite rarely and were accounted for in the partitioning of pan-genome portions. The large sample size was able to recapitulate the key genes due to large sample size.

**Structural protein analysis of identified AMR genes**. For identified AMR genes, the *ssbio* software was used to gain gene-specific, protein sequence and structure based information about residue-level changes (SNPs and deletions) present in the *M. tuberculosis* alleles[19]. Each AMR gene was mapped to a reference protein sequence file obtained from UniProt[39] and sequence-based metadata identifying protein-specific features (e.g., active sites, secondary structures, and mutations in studied wild-type strains) was used to determine the occurrence of allele-specific AMR mutations within the gene feature set (Supplementary Data 6). When available, AMR genes were additionally mapped to experimentally obtained protein structures from the RCSB Protein Data Bank or to homology structures generated using the Iterative Threading ASSEmbly Refinement (I-TASSER) platform[40,41]. To help elucidate the mechanistic effects of AMR mutations, both AMR mutations and the residue-level feature set were mapped to these structures and visualized using the NGLview Jupyter notebook plugin[42]. The structural information was utilized to calculate distances between each mutation and annotated protein feature (Supplementary Data 6).

**Code availability**. The computational platform is provided as a github code repository.

## Data availability

All data utilized in this study is publicly available at the PATRIC database. Identifiers for the 1595 genomes are provided in the Supplementary Information (Supplementary Data 7). References for the published and unpublished data sets can be found in the Supplementary Information (Supplementary Data 7). The sequencing data for the TB Antibiotic Resistance Catalog (TB-ARC) projects (Supplementary Data 7) were generated at the Broad institute. Additional information for each of these unpublished projects can be found at the Broad Institute website (https://olive.broadinstitute.org/projects/tb_arc).

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

## Acknowledgments

We thank Anand Sastry for helpful discussions regarding machine learning. This research was supported by the NIH NIAID grant (1-U01-AI124316-01), and the NIH NIGMS (award U01GM102098).

## Author contributions

E.K., J.M.M. and B.O.P. conceived and designed the study. E.K. conducted all analysis, with contributions from E.C., N.M., D.H., and J.M.M., E.K., Y.S. and J.M.M. performed the pan-genome analysis. E.K. and D.H. performed the epistatic interaction analysis. E.C. and N.M. developed the 3D protein structural analysis pipeline. E.K., J.T.Y., E.C., N.M., Y.S., N.D., A.A., L.Y., D.H., V.N., J.M.M. and B.O.P. provided study oversight, wrote the manuscript, and edited the manuscript. J.M.M. and B.O.P. managed the study. All authors reviewed and approved the final manuscript.

## Additional information

**Competing interests:** The authors declare no competing interests.

