## [Peer Review file · Nature Communications]

Reviewers' comments:

Reviewer #1 (Remarks to the Author):

Review of "Machine learning and structural analysis of Mycobacterium tuberculosis pan-genome identifies genetic signatures of antibiotic resistance"

Summary

Kavvas et al have analysed 1595 *M. tuberculosis* genomes from the PATRIC database, along with phenotypic data (drug resistance), and sought to find mutations which cause drug resistance, and potential epistatic interactions. They start by ignoring the non-coding genome, and then build an allelic representation of all the gene sequence present in these samples. After decomposing this sequence into distinct chunks, (this can be seen as similar to that seen in graph representations of pan-genomes, such as Jaillard et al (<https://www.biorxiv.org/content/early/2017/03/03/113563>)), they use mutual information between presence-absence vectors of sequence markers and antibiotic resistance profiles to search for antibiotic resistance genes. This works strikingly well, finding the known genes *rpoB*, *pncA*, *embB*, *rpsL*, *katG* and *gyrA* - we return to this below. The authors then go on to use support vector machines to discover more associations, and then map their findings onto structures where possible. As an aside, they also report some analyses of the pan-genome of *M. tuberculosis* which perhaps is not that relevant to this paper.

This is an interesting paper which performs novel and interesting analyses which are a contribution to the community - I think it should be published, but there are a few issues I would like to resolve first.

Major issues

1. Drug resistance in *M. tuberculosis* has evolved and spread in response to our use of anti-tubercular drugs. Ever since drug resistance first started to evolve during clinical trials, treatment has involved multiple drugs. Current best practise involves four first-line drugs as a standard regiment - isoniazid, ethambutol, pyrazinamide and rifampicin. As a result, samples are rarely exposed to just one drug, and drug resistances tend to co-occur. It is therefore difficult a priori (for me) to see how a purely genotype and phenotype -driven method can possibly tell which gene is associated with which drug. Indeed the authors mention that the dataset is 59% resistant to Rifampicin and 59% resistant to Isoniazid (I assume there is a huge overlap between these sets), so it might be hard to tell which gene associates with which drug. Indeed, if you look at Figure 2, the highlighted peak for isoniazid is in *katG*, and presumably is the well-known single dominant mutation - but there is a much higher peak to the left, which is unlabelled, and is very close to a labelled *rpoB* allele. Is this in fact an example of this issue? Furthermore, the highlighted *rpoB* allele is clearly also associated with peaks for pyrazinamide, streptomycin, isoniazid resistance also. I don't think this is a failing in the paper, but it would be good to be clearer. The MI method is essentially looking for association with AMR profiles, and it would be interesting to see if there is a systematic way to deconvolve out an association of a region (gene) with a drug.

2. I would like to understand how we distinguish epistasis, as discovered by the method in this paper,

from lineage effects (see Earle et al, mentioned above for more discussion). Any allele in tight linkage with an AMR allele will have equivalent signal by the method in this paper (Kavvas et al I mean, not Earle et al), and so we would expect to see association with other alleles, across the genome, that happen to be in linkage with the AMR gene (ie on the same branch of the phylogeny). Are the authors effectively relying on homoplasy to remove this confounding?

3. I am not very convinced at all about the mapping onto structures section of the paper. The authors seem to be drawing confidence from finding mutations are either close to the binding site or located in a domain that appears biochemically important - however there are lots of examples where that may well be so and the mutation has no effect, and contrary examples where the mutation is miles away but is resistant.

4. The section on the pan-genome being open is not really relevant to this paper. It is already well known that there is little if any horizontal gene transfer into *M. tuberculosis*; the signal seen in the PE/PPE genes seems to me likely to be due to non-allelic recombination events leading to diversity. I think it is interesting that this much diversity is being seen, but I don't think most people would see this as causing an open pan-genome, as they are not novel genes with new functions (eg compare var genes in *P. falciparum*). The way alleles have been defined is nice, and useful, but the pan-genome open-ness feels to me like a tangent. I'm not explicitly asking for it to be removed, but if in revision the authors need space to give better exposition for something else, to my mind losing the open pan-genome stuff would be no big loss to this paper (although I would like it published so I could reference it!)

Other issues

1. On line 135 the authors mention that their results show that exact sequence match and allele frequencies alone are enough to identify AMR genes. This is correct, but also well known from the various methods that use K-mer presence to detect causal alleles - for example Earle et al (<https://www.ncbi.nlm.nih.gov/pmc/articles/PMC5049680/>), Jaillard et al mentioned above, and Lees et al (<https://www.nature.com/articles/ncomms12797>)

2. Compensatory mutations were mentioned, but not the well known rpoC compensatory mutation for rpoB resistance - is this maybe there in the data but not highlighted?

Reviewer #2 (Remarks to the Author):

To be best of my knowledge this is the first time Support Vector Machines are used to identify drug resistance genes in *Mtb* and epistatic interactions.

Although I understand that wet-lab experimental validation of the candidate loci may be outside the scope of this study, a biological interpretation and mechanistic insights about the role of novel genes and epistatic interactions is currently lacking.

Find below a list of necessary changes before this work can be published:

- Table 1. There are only six genes with a superscript 1 in Table 1, which does not add to the

“additional seven known AMR gene-antibiotic relations”

- Supplementary Data File 2. The gene_name column in sheets 2-11 does not always contain the expected gene name for its corresponding Rv name, which makes interpretation difficult. For instance, the locus RV1908C should have 'katG' as its gene name and Rv2043c should have 'pncA'.

- I find striking that RV2043C/pncA is labelled as an accessory gene (Supplementary Data File 2, pyrazinamide_SVM_SGD sheet). Have the authors check if Mtb strains lacking pncA or with truncated versions of it are pyrazinamide resistant? Loss of PZase enzymatic activity encoded by pncA is the most frequent mechanisms of pyrazinamide resistance.

Line 155. “SVM method revealed an abundance of AMR-implicated genes involved in metabolic pathways”. The authors should quantify what they mean by “abundance” here, for example as a percentage, so that this value can be compared to that of the 73% among known AMR-determinants.

Line 158. “We found over 30 genes related to cell wall processes”. The authors should provide a percentage here.

Figure 3. Both in Figure 3 and throughout the text the authors should refer to the actual genetic variants (annotated as amino acid changes) as opposed to numbered alleles only. Otherwise, the readers won't be able to identify what mutations are found to be associated with drug resistance and those involved in epistatic interactions.

Line 214. The authors state that “embR alleles appeared sparsely across the ensemble of SVM hyperplanes”. This statement uses very technical terms and it is hard to interpret. Is this the result of embR alleles being rare in the population?

Line 223. The authors state that “resistant-dominant ubiA alleles (i.e., those with high positive LOR), 1 and 3, occurred exclusively in the background of non-susceptible-dominant embR alleles”, however, in Figure 3a, it looks like ubiA alleles 2 and 4 are the ones co-occurring with embR alleles 2 and 4. Please check.

Line 226. The authors state that “Furthermore, neither embR nor ubiA were significantly associated with ethambutol AMR in pairwise associations tests”. Can the authors explain why these two genes were not identified in pairwise association tests? Is the because of a lower allele frequency of mutations in these two genes in the population as opposed to common embB mutations?

Supplementary Data File 4 contains a lot of different co-occurrence tables. Numbered alleles should be annotated to represent actual genetic variants and amino acid changes so that these results can be compared to previous studies.

Line 244. It is not clear to me what the authors mean by “the variation in AMR phenotypes across the different alleles were determined to be significant by the machine learning algorithm and thus motivated further investigation.”.

Section “Machine learning uncovers genetic interactions contributing to AMR”. The strictly clonal population structure of Mtb, which is divided into seven lineages and these further split into sub-lineages, may have confounded the epistatic analysis. The authors should make sure that co-occurrence of alleles from different genes among resistant strains happens across different genetic backgrounds (i.e. lineage/sub-lineages) to provide further evidence of this putative epistatic interaction. For instance, in Figure 3b, katG allele 3 co-occurs frequently with oxcA alleles 2, 3, 6 and

7; do the 78 strains containing both katG allele 3 and oxcA allele 2 belong to the same Mtb sub-lineage? If so, this putative epistatic interaction may be the result of linkage (hitch-hiking) of these mutations in the same clonal background and therefore a likely false positive. If these 78 strains belong to different genetic backgrounds (i.e. lineage/sub-lineages), then this provides evidence of independence acquisition of these mutations. The authors can make use of existing SNP typing schemes (Coll et al. 2014) and/or construct a robust whole-genome phylogeny (See comment below).

Methods:

Line 545. *M. tuberculosis* strain dataset. The authors state that they “selected a representative set of 1,595 *M. tuberculosis* strains for which AMR testing data was available from the PATRIC database”. On the 6th of March 2018, the PATRIC database has 9,314 available *M. tuberculosis* genomes (https://patricbrc.org/view/Taxonomy/1773#view_tab=genomes). The authors should at least prepare a supplementary data file with information on the 1,595 used in this study, including their ENA or Genbank accession numbers; and cite not only the PATRIC database but the publication where isolates were originally sequenced and presented. The authors should note that they should not make use of unpublished genomes.

Line 554. In addition to the use of CD-hit package, the authors may want to use another established bioinformatics tool to characterise bacterial pan-genomes: (Page et al. 2015). If both CD-hit and Roary yield similar results, the readers will be more confident about the pan-genome analysis.

Line 571. The phylogenetic approach used by the authors is not valid. Given that the whole-genome sequences of the 1,603 strains are available, the authors must make use of SNPs at the core genome to create a whole-genome phylogeny that is comparable to that already published ((Coll et al. 2014), (Comas et al. 2013)). Selection of just a few housekeeping genes is not enough to obtain a high-resolution phylogeny required for this type of genomic studies. Furthermore, four of the seven housekeeping genes chosen (rpoB, katG, gyrA and rpoC) are likely to contain homoplastic SNPs among resistant strains that will inevitably distort the phylogeny.

Line 589. The authors state that “*M. tuberculosis* has an open pan-genome.” The authors also note that “a significant portion of the unique and accessory genome was attributed to Pro-Glu (PE)-related proteins and hypothetical proteins (Fig. 1b).” These results could be the result of assembly and/or gene annotation artefacts. PE/PPE genes have a high GC content, which results in lower depth of coverage in these regions, and some members of this family of genes are highly repetitive, which make them inherently difficult to assemble from short sequence reads generated by Illumina sequencers. Fragmented assemblies at the boundaries of PE/PPE genes might lead to a higher number of predicted PE/PPE clusters. In order to rule out the possibility of assembly artefacts confounding the pan-genome analysis, I would recommend the authors repeat the pan-genome analysis using the subset of ‘Complete’ *Mtb* genomes sequenced using long-read sequencing (PacBio) and compare their results with those of the whole dataset. The authors should also measure the length of newly identified genes as they add more strains. Are newly identified genes shorter than core-genome genes? If so, they could be the result of miss-assemblies. The other necessary check is to quantify the amount of new DNA sequences (in base pairs) as more strains are added to make sure that the increasing number of new genes observed matches with an expected increase in new DNA sequences. The authors should perform these extra analyses before stating that “*M. tuberculosis* has an open pan-genome.” as it is established in the field that *Mtb* has a closed pan-genome due to the lack of inter-strain recombination.

Coll F, McNerney R, Guerra-Assunção JA, Glynn JR, Perdigão J, Viveiros M, Portugal I, Pain A, Martin N, Clark TG. 2014. A robust SNP barcode for typing Mycobacterium tuberculosis complex strains. *Nature Communications* 5: 4812.

Comas I, Coscolla M, Luo T, Borrell S, Holt KE, Kato-Maeda M, Parkhill J, Malla B, Berg S, Thwaites G, et al. 2013. Out-of-Africa migration and Neolithic coexpansion of Mycobacterium tuberculosis with modern humans. *Nature Genetics* 45: 1176–1182.

Page AJ, Cummins CA, Hunt M, Wong VK, Reuter S, Holden MTG, Fookes M, Falush D, Keane JA, Parkhill J. 2015. Roary: rapid large-scale prokaryote pan genome analysis. *Bioinformatics* 31: 3691–3693.

Reviewer #3 (Remarks to the Author):

The work targets an interesting and hot topic that is the investigation of AMR characteristics and evolution in MTB. To come to the conclusions the authors use statistical methods as well as machine learning approaches. Doing so the authors identified known resistance markers as well as 24 new genes somehow associated with drug resistance. A really interesting part is the analysis of interactions between genes, which is a unique feature of this study. Additionally, the authors prevent themselves of statistical artifacts by investigating the crystal structure of the newly identified genes and how those genes could play a role in AMR.

The paper is well written and the results are clearly presented.

The study has four main conclusions and thus the article has some length in reading. There is so many really interesting stuff included and each step make sense, but would like a shorter version focusing on the main points of the study. The sample size and selection of data is reasonable and leads to a representative dataset. I wonder about the term “platform”, is the code somewhere provided or is there a platform where you can upload your own data and get a classification?

I would suggest to make the MLST scheme with more than just 7 house keeping genes.

In Table 1 it is a profit of the SVM compared to the ANOVA and other methods, but to be honest when inhA would not be associated with INH resistance the method needs definitely a reconstruction.

The key points of the study needs to be more precisely described e.g. focusing on only 1-2 major points and skipping for example the county distribution.

This font color represents our comment to reviewers

This font color represents text in the main manuscript or supplementary that has been changed.

This highlight represents the text that has been changed in the main manuscript or supplementary text.

Reviewers' comments:

Reviewer #1 (Remarks to the Author):

Review of "Machine learning and structural analysis of Mycobacterium tuberculosis pan-genome identifies genetic signatures of antibiotic resistance"

Summary

Kavvas et al have analysed 1595 M. tuberculosis genomes from the PATRIC database, along with phenotypic data (drug resistance), and sought to find mutations which cause drug resistance, and potential epistatic interactions. They start by ignoring the non-coding genome, and then build an allelic representation of all the gene sequence present in these samples. After decomposing this sequence into distinct chunks, (this can be seen as similar to that seen in graph representations of pan-genomes, such as Jaillard et al (<https://www.biorxiv.org/content/early/2017/03/03/113563>)), they use mutual information between presence-absence vectors of sequence markers and antibiotic resistance profiles to search for antibiotic resistance genes. This works strikingly well, finding the known genes rpoB, pncA, embB, rpsL, katG and gyrA - we return to this below. The authors then go on to use support vector machines to discover more associations, and then map their findings onto structures where possible. As an aside, they also report some analyses of the pan-genome of M. tuberculosis which perhaps is not that relevant to this paper.

This is an interesting paper which performs novel and interesting analyses which are a contribution to the community - I think it should be published, but there are a few issues I would like to resolve first.

Major issues

1. Drug resistance in M. tuberculosis has evolved and spread in response to our use of anti-tubercular drugs. Ever since drug resistance first started to evolve during clinical trials, treatment has involved multiple drugs. Current best practise involves four first-line drugs as a standard regiment - isoniazid, ethambutol, pyrazinamide and rifampicin. As a result, samples are rarely exposed to just one drug, and drug resistances tend to co-occur. It is therefore difficult a priori (for me) to see how a purely genotype and phenotype -driven method can possibly tell which gene is associated with which drug. Indeed the authors mention that the dataset is 59% resistant to Rifampicin and 59% resistant to Isoniazid (I assume there is a huge overlap between these sets), so it might be hard to tell which gene associates with which drug. Indeed, if you look at Figure 2, the highlighted peak for isoniazid is in katG, and presumably is the well-known single dominant mutation - but there is a much higher peak to the left, which is unlabelled, and is very close to a labelled rpoB allele. Is this in fact an example of this issue? Furthermore, the highlighted rpoB allele is clearly also associated with peaks for pyrazinamide, streptomycin, isoniazid resistance also. I don't think this is a failing in the paper, but it would be good to be clearer. The MI method is essentially looking for association with AMR profiles, and it would be

interesting to see if there is a systematic way to deconvolve out an association of a region (gene) with a drug.

The reviewer makes an interesting observation in Figure 2 that there are multiple large peaks found in a specific antibiotic. The incidence of multiple AMR peaks is indeed an example of the simultaneous use of multiple antibiotics in TB treatment. This feature was a key driver for our chosen visualization of allele-AMR associations portrayed in Figure 2 (now Figure 1). The discussion of this issue is located in the supplementary text section entitled “**Motivation for using mutual information and observation of shared AMR signals across multiple antibiotics**”. We agree that it would be interesting and valuable to deconvolve out an association of a region with a drug. Our study, however, does not provide a solution to this problem. Future efforts may similarly take inspiration from information theory and apply methods such as independent component analysis or the information bottleneck method ¹ towards deducing independent signals corresponding to unique antibiotics.

2. I would like to understand how we distinguish epistasis, as discovered by the method in this paper, from lineage effects (see Earle et al, mentioned above for more discussion). Any allele in tight linkage with an AMR allele will have equivalent signal by the method in this paper (Kavvas et al I mean, not Earle et al), and so we would expect to see association with other alleles, across the genome, that happen to be in linkage with the AMR gene (ie on the same branch of the phylogeny). Are the authors effectively relying on homoplasy to remove this confounding?

While it is possible that lineage effects confound the results obtained by our pairwise association calculations, they are unlikely to confound the results obtained by our machine learning approach and subsequent epistasis calculations (the machine learning provides the set of gene-gene interactions to test for epistasis). This is because, in the case of pairwise associations, each allele is viewed independently of one another. Therefore, an allele in tight linkage with the key AMR allele will appear with similar significance through pairwise-associations. In the case of our SVM approach, however, an allele in tight linkage with the major AMR allele will not be identified (i.e., incorporated as a variable in the SVM classifier function) unless the allele in tight linkage provides further predictive accuracy. Our utilization of an L1-norm in the SVM limits the number of alleles that can be used to classify the AMR phenotypes, thereby eliminating alleles that are redundant (i.e., hitchhikers). We should clarify that the set of tested epistatic interactions come from correlations amongst the ensemble of SVM classifier functions (Supplementary Figure 5, shown below), not correlations amongst allele frequencies. Thus, the identified epistatic interactions are not prone to lineage effects because the set of tested gene-gene interactions are selected through a machine learning approach that negates redundant and non-predictive features.

To evaluate whether lineage effects confound our resulting epistatic interactions, we first categorized our strains into lineages using a SNP-barcode² and then viewed the distribution of these lineages within the epistatic interactions. We find that while homoplasmy is certainly a strong determinant, which is known to be a key feature of *M. tuberculosis* AMR evolution^{3,4}, our identified set of epistatic interactions extend into multiple lineages, which shows that alleles in tight linkage do not confound our machine learning-driven approach. For example, the strains containing the co-occurrence of *katG* allele 3 and *oxcA* allele 2 (shown in Fig 3) are spread across lineages 1, 2, and 3 (see barplot below).

The approach taken by Earle et al. is indeed fascinating and shows that the combination of Principal Component Analysis (PCA) and Linear Mixture Model (LMM) accounts for lineage confounding effects. The authors find that the PCA decomposition of the k-mer variant matrix returns basis vectors that correspond to lineages. We thank the reviewer for drawing our attention to this work and feel that our study provides additional insights to the study of linkage effects and AMR.

- I am not very convinced at all about the mapping onto structures section of the paper. The authors seem to be drawing confidence from finding mutations are either close to the binding site

or located in a domain that appears biochemically important - however there are lots of examples where that may well be so and the mutation has no effect, and contrary examples where the mutation is miles away but is resistant.

The structural mutation mapping does indeed lack both quantitative rigor and a biological basis in providing confidence to the selected AMR alleles. As the reviewer notes, the incidence of both a resistance-dominant mutation and an annotated structural region does not necessarily support the conclusion that the resistance-dominant mutation is *more likely* to be casual. In order to address this incongruence, we have softened the claims made by our structural analysis and have restricted the insights to potential mechanistic insights underlying the selection of the alleles. While the incidence of mutations and structural features does not provide confidence in the causality of the allele, the incidence of features may provide insights into the mechanistic driver of selection. Therefore, the structural analysis solely enables the inference of potential mechanistic effects driving the selection of the alleles. In our study, we provided control cases (i.e., *inhA*, *katG*) in which the incidence of mutations with alleles recapitulated known mechanism-of-action (MoA). Structural analysis thus enabled deeper hypothesis generation regarding the MoA. We do not claim that it added any more support to the causality of an implicated allele. Our goal was instead to go beyond the reporting of alleles by trying to link them to MoA, we feel that this process has helped to rank mutations of interest for future validation studies.

Structural analysis of implicated AMR genes suggest a mechanistic driver of selection

4. The section on the pan-genome being open is not really relevant to this paper. It is already well known that there is little if any horizontal gene transfer into *M. tuberculosis*; the signal seen in the PE/PPE genes seems to me likely to be due to non-allelic recombination events leading to diversity. I think it is interesting that this much diversity is being seen, but I don't think most people would see this as causing an open pan-genome, as they are not novel genes with new functions (eg compare var genes in *P. falciparum*). The way alleles have been defined is nice, and useful, but the pan-genome open-ness feels to me like a tangent. I'm not explicitly asking for it to be removed, but if in revision the authors need space to give better exposition for something else, to my mind losing the open pan-genome stuff would be no big loss to this paper (although I would like it published so I could reference it!)

We thank the reviewer for providing comments on the pan-genome analysis portion of this study. As the reviewer points out, the pan-genome provided the foundation for our perspective of genetic variation (i.e., our reference-agnostic exact allele view), which is a unique and an important feature of our study. We agree with the reviewers, however, that our pan-genome analysis --- which describes the shape of the pan-genome, distribution of virulence factors, and counteractome genes --- is certainly tangential to the primary goal of identifying AMR genes by machine learning. Therefore, we have moved the analysis portions of the pan-genome section to the supplementary material.

Furthermore, we have recomputed the shape of the pan-genome by filtering out PE/PPE genes and genes with lengths that were significantly longer (>1 standard deviation) than the mean gene length of 1000 bp which are likely result of sequencing or annotation errors. In total, this led to the removal of 1,335 genes clusters from the pan-genome. The majority of these genes (826) were PE/PPE genes.

Following the removal of these genes we find that the pan-genome is indeed closed for our 1595 strains of *M. tuberculosis*, see below:

Other issues

1. On line 135 the authors mention that their results show that exact sequence match and allele frequencies alone are enough to identify AMR genes. This is correct, but also well known from the various methods that use K-mer presence to detect causal alleles - for example Earle et al (<https://www.ncbi.nlm.nih.gov/pmc/articles/PMC5049680/>), Jaillard et al mentioned above, and Lees et al (<https://www.nature.com/articles/ncomms12797>)

We thank the reviewer for drawing our attention to the various methods that used K-mer presence to detect causal alleles. While our approach is not quite the same as K-mer approach (full exact sequence vs k-mer), the concept is similar in its word-based perspective and certainly should be cited in our study. We have now included citations for these papers in our study at relevant point.

These results suggest that allele frequencies based on exact sequence (i.e., without a metric for genetic distance) are capable of identifying AMR genes, which has previously been shown with k-mer based approaches^{5 6 7}.

2. Compensatory mutations were mentioned, but not the well known *rpoC* compensatory mutation for *rpoB* resistance - is this maybe there in the data but not highlighted?

We thank the reviewer for highlighting a key false negative in our gene-gene epistasis calculations, which has led us to an improvement in our gene-gene epistasis workflow. The *rpoC* mutations were indeed identified as significant by our SVM approach (see **Supplementary Data File 2, sheet "rifampicin_SVM_SGD"**). The *rpoC* mutations, however, were previously not found to have significant epistatic interactions with *rpoB* by our quantitative epistasis analysis and highlighted a flaw in our logistic regression gene-gene epistasis calculations. Upon second review of our methodology, we noticed that low signal genes uncovered by the ensemble SVM were being tested in these gene-gene epistasis tests.

For example, the *Rv2090-lpqZ* interaction highlighted in Figure 4 was previously tested to be significant despite *Rv2090* appearing with a low signal in the SVM ensemble. While the gene-gene co-occurrence table is interesting, the interaction is not more significant than the *rpoB-rpoC* interaction.

Our updated epistasis calculations now identify *rpoB-rpoC* as a significant gene-gene epistatic interaction by limiting the tested interactions to those with a high correlation amongst the top 25 weighted alleles (shown below, and provided in Supplementary Data File 4). We have updated both Supplementary Table 3 and Supplementary Data File 4 with the results generated by the improved epistasis methodology.

Furthermore, we have now updated Figure 3 (shown below) by removing the *lpqZ-Rv2090* co-occurrence table (part C).

The methods section has now been updated to reflect the improvement in methodology.

We selected the top 100 gene-gene correlations that occur amongst the top 25 ranked SVM alleles for each antibiotic. We limited the correlations to the top 25 ranked alleles in order to avoid the case when low weighted alleles appear sparsely with other low weighted alleles which lead to significant correlations.

Reviewer #2 (Remarks to the Author):

To be best of my knowledge this is the first time Support Vector Machines are used to identify drug resistance genes in Mtb and epistatic interactions.

Although I understand that wet-lab experimental validation of the candidate loci may be outside the scope of this study, a biological interpretation and mechanistic insights about the role of novel genes and epistatic interactions is currently lacking.

We thank the reviewer for appreciating the novelty our work. We agree that wet-lab validation is outside the scope of this study but hope that our work will guide future validation experiments. Due to space constraints we were forced to place much of the biological interpretation and mechanistic insights about the role of novel genes and epistatic interactions in the Supplementary Note. In particular, we describe identified toxins within the context of literature (see section titled "Adaptations in toxins are associated with XDR in *M. tuberculosis*"), how the genetic background of *oxcA* resistant-dominant alleles may guide

experimentation (see section titled “*Epistatic and protein-structure-guided generation of experimental hypothesis*”), and suggest relationships between implicated genes and potential excessive antibiotic treatment using geographic data (see section titled “*Geographic contextualization suggests modulation of antibiotic treatment*”). We provided this information when both literature and structural insights were available. In cases when such details were unknown (i.e., Rv3471c) --- which is the majority of the implicated genes --- we abstained from drawing weak hypothesis and made note of the key details for the reader in Table 2 (e.g., Rv3471c is predictive of ETA, XDR, and SM AMR phenotypes, resistant dominant allele (48/50) contains an SNP inside the Cupin 1 domain, and appears in South Africa). While providing biological interpretations for the implicated genes is tempting, statistical models and subsequent analysis are fundamentally limited in the conclusions they can draw. Therefore, we stayed away from providing further information that may bias the interpretation of the implicated AMR genes, and focused our discussion on the computational platform and the mathematical outputs that it provides. We hope our work and this discussion will provide hypothesis and guidance for future validation studies.

Major Issues

Find below a list of necessary changes before this work can be published:

- Table 1. There are only six genes with a superscript 1 in Table 1, which does not add to the “additional seven known AMR gene-antibiotic relations”

We had incorrectly placed an asterisk next to the *embR* gene instead of a “1”. The *embR* alleles were not associated with ethambutol in the pairwise association tests (i.e., GWAS). Furthermore, since the numeric value of 1 may be confusing along with the citation number, we have replaced the “1” superscript with an asterisk “*”. We have now made these corrections and the total number of “*” superscripts now adds up to 7, as mentioned in the paper. Note that *inhA* is counted twice since it did not appear in both isoniazid and ethionamide pairwise-association tests (*inhA* mutations are known to confer resistance to both drugs).

- Supplementary Data File 2. The gene_name column in sheets 2-11 does not always contain the expected gene name for its corresponding Rv name, which makes interpretation difficult. For instance, the locus Rv1908C should have ‘katG’ as its gene name and Rv2043c should have ‘pncA’.

We thank the reviewer for pointing this out. We have now included correct id to name mappings under the “gene_name” column for all genes in which a name is provided. We utilized the mappings provided in the Mycobrowser database for *Mycobacterium tuberculosis* H37Rv⁸.

- I find striking that RV2043C/pncA is labelled as an accessory gene (Supplementary Data File 2, pyrazinamide_SVM_SGD sheet). Have the authors check if Mtb strains lacking pncA or with truncated versions of it are pyrazinamide resistant? Loss of PZase enzymatic activity encoded by pncA is the most frequent mechanisms of pyrazinamide resistance.

Out of the 312 strains lacking the pan-genome cluster corresponding to *pncA* (Cluster 3930), only 53 strains were tested for pyrazinamide susceptibility. Of these 53 strains, 51 were resistance (96%) and 2 were susceptible (4%). This is precisely in line with what the reviewer mentions as the most frequent mechanism of pyrazinamide resistance and therefore provides a reason for the partitioning of *pncA* into the accessory genome.

Line 155. “SVM method revealed an abundance of AMR-implicated genes involved in metabolic pathways”. The authors should quantify what they mean by “abundance” here, for example as a percentage, so that this value can be compared to that of the 73% among known AMR-determinants.

Thank you for this comment. Out of the 472 implicated AMR genes described across the 10 AMR classifications, 172 were annotated as metabolic by COG, (172/472, 37%).

Line 158. “We found over 30 genes related to cell wall processes”. The authors should provide a percentage here.

Out of the 472 implicated AMR genes described across the 10 AMR classifications, 36 were annotated as cell wall/membrane/envelope biogenesis by COG (36/472, 8%). We have made these corrections in the main text.

The SVM method revealed an abundance of AMR-implicated genes involved in metabolic pathways (119/317, 37.5%) (Supplementary Data File 2). In fact, the majority of the known AMR-determinants are metabolic enzymes (24/33, 73%). We found over 20 genes related to cell wall processes (26/317, 8.2%), which is consistent with previous findings of convergent AMR evolution in *M. tuberculosis*³.

Figure 3. Both in Figure 3 and throughout the text the authors should refer to the actual genetic variants (annotated as amino acid changes) as opposed to numbered alleles only. Otherwise, the readers won't be able to identify what mutations are found to be associated with drug resistance and those involved in epistatic interactions.

Instead of defining genetic features as amino acid changes relative to the H37Rv reference sequence, we take a reference agnostic approach by representing each allele as a unique sequence variant. Our view of genetic variation, however, makes connecting the genetic variants to amino acid changes challenging as a genetic variant may have numerous amino acid changes, including deletions, insertions etc. For example, there are 3 distinct polymorphisms found in *oxcA* allele 3 (see Figure 4 Main text, now Figure 3), which makes it rather challenging to portray within the space constraints of an allele co-occurrence table, or in the main text.

While our analysis is agnostic to a single reference sequence (with exception to the protein structural analysis), we agree that the readers should be able to relate the numbered alleles to specific mutations. Therefore, we have provided a mutational mapping of all alleles in the predicted AMR genes, defined relative to the H37Rv reference strain (see **Supplementary Table 4, column “mutation_residues”**). Furthermore, metadata from uniprot describing previous annotations of mutations is additionally provided (see **Supplementary Table 4, column “uniprot_features”**), which should enable researchers to differentiate between previously known mutations and potential new ones. We believe that this allows for comparison of our work with other studies, while also maintaining our perspective of genetic variation taken in this study.

Line 214. The authors state that “embR alleles appeared sparsely across the ensemble of SVM hyperplanes”. This statement uses very technical terms and it is hard to interpret. Is this the result of embR alleles being rare in the population?

We agree that the statement is too technical and should be simpler to interpret. We have now altered this line to read as follows,

“Although the *embR* alleles appeared few times across the multiple SVM simulations, their appearance was highly correlated with alterations in the sign and weight of the *ubiA* allele (see Supplementary Figure 6). This implies that *embR* is only a predictive feature within the context of *ubiA*, which may result from the weak penetrance of *embR* alleles within *M. tuberculosis* (see Figure 2a)”

Interestingly, the *embR* alleles are neither rare nor highly resistant-dominant (see Figure 3a). Therefore, it's not entirely surprising that the *embR* allele does not appear in pairwise association tests (now corrected in Table 1). The observation that *embR* alleles appeared sparsely across the ensemble of SVM hyperplanes implies that the effect of *embR* is subtle, or hidden, and becomes important (i.e., large weighting on SVM hyperplane) when appearing along with another allele and specific weighting of that allele. We provided a figure describing this sparse behavior (Supplementary Material File 3, “ethambutol_SVM_SGD_iterations.png”). The ability to uncover these features is the primary value of a machine learning approach.

Line 223. The authors state that “resistant-dominant *ubiA* alleles (i.e., those with high positive LOR), 1 and 3, occurred exclusively in the background of non-susceptible-dominant *embR* alleles”, however, in Figure 3a, it looks like *ubiA* alleles 2 and 4 are the ones co-occurring with *embR* alleles 2 and 4. Please check.

The reviewer is correct. *ubiA* alleles 2 and 4 are the resistant-dominant alleles co-occurring with *embR* alleles 2 and 4. We have resolved this error in the revised manuscript.

We observed that the resistant-dominant *ubiA* alleles (i.e., those with high positive LOR), 2 and 4, occurred exclusively in the background of non-susceptible-dominant *embR* alleles (Fig. 3a).

Line 226. The authors state that “Furthermore, neither *embR* nor *ubiA* were significantly associated with ethambutol AMR in pairwise associations tests”. Can the authors explain why these two genes were not identified in pairwise association tests? Is the because of a lower allele frequency of mutations in these two genes in the population as opposed to common *embB* mutations?

The *ubiA* alleles were not identified in the AMR pairwise association tests due to having a low resistance-conferring allele frequency, as seen in Figure 3b. The *embR* alleles similarly lacked a significant association to the AMR phenotypes with respect to other alleles. In both cases, this lack of significant

AMR association in *ubiA* and *embR* alleles is primarily due to being clouded by other alleles that are in close linkage with the key resistance-conferring allele (i.e., lineage effects, hitchhiker mutations, etc). Specifically, by “clouded”, we mean that these other alleles will have a higher association than *ubiA* and *embR*, which in some sense pushes *ubiA* and *embR* off the list of implicated genes.

Supplementary Data File 4 contains a lot of different co-occurrence tables. Numbered alleles should be annotated to represent actual genetic variants and amino acid changes so that these results can be compared to previous studies.

See response above to reviewer comment about Figure 3. Because we take a reference-agnostic approach, listing specific amino-acid changes is challenging. Instead we provide a table with all variants explicitly laid out in Supplementary Table 4, column “mutation_residues”.

Line 244. It is not clear to me what the authors mean by “the variation in AMR phenotypes across the different alleles were determined to be significant by the machine learning algorithm and thus motivated further investigation.”.

We agree that this sentence is ambiguous. We have now rewritten this sentence as follows.

“The implicated alleles were identified by the machine learning algorithm as predictive features for classifying AMR phenotypes and thus motivated further investigation”.

Section “Machine learning uncovers genetic interactions contributing to AMR”. The strictly clonal population structure of Mtb, which is divided into seven lineages and these further split into sub-lineages, may have confounded the epistatic analysis. The authors should make sure that co-occurrence of alleles from different genes among resistant strains happens across different genetic backgrounds (i.e. lineage/sub-lineages) to provide further evidence of this putative epistatic interaction. For instance, in Figure 3b, *katG* allele 3 co-occurs frequently with *oxcA* alleles 2, 3, 6 and 7; do the 78 strains containing both *katG* allele 3 and *oxcA* allele 2 belong to the same Mtb sub-lineage? If so, this putative epistatic interaction may be the result of linkage (hitch-hiking) of these mutations in the same clonal background and therefore a likely false positive. If these 78 strains belong to different genetic backgrounds (i.e. lineage/sub-lineages), then this provides evidence of independence acquisition of these mutations. The authors can make use of existing SNP typing schemes (Coll et al. 2014) and/or construct a robust whole-genome phylogeny (See comment below).

We thank the reviewer for linking existing SNP types schemes. We have now built a new phylogenetic tree of our strains using existing SNP typing schemes ² (see comment below in Methods related to our previous phylogenetic approach). Using our updated phylogenetic tree, we investigated the location of the 78 strains containing both *katG* allele 3 and *oxcA* allele 2 and found that these strains are spread across lineages 1, 2, and 3 (**see barplot below**). Of the 78 strains, 60 were contained in lineage 2.2 with 30 further categorized into sublineage 2.2.1. Of the remaining 18 strains, 5 were classified as lineage 1 and 13 were classified as lineage 3. While the co-occurrence of these two alleles (*katG* allele 3 and *oxcA* allele 2) in different lineages indicates that these putative epistatic interactions are likely not the result of linkage, we should note that interacting mutations are known to appear in clonal backgrounds of *M. tuberculosis* ⁴ and are descriptive of convergent evolution ³.

Methods:

Line 545. *M. tuberculosis* strain dataset. The authors state that they “selected a representative set of 1,595 *M. tuberculosis* strains for which AMR testing data was available from the PATRIC database”. On the 6th of March 2018, the PATRIC database has 9,314 available *M. tuberculosis* genomes (https://patricbrc.org/view/Taxonomy/1773#view_tab=genomes). The authors should at least prepare a supplementary data file with information on the 1,595 used in this study, including their ENA or Genbank accession numbers; and cite not only the PATRIC database but the publication where isolates were originally sequenced and presented. The authors should note that they should not make use of unpublished genomes.

We have now included a supplementary table that includes GenBank accession numbers for the genomes, references, sequencing details, and other metadata for the selected 1,595 *M. tuberculosis* strains (see **Supplementary Table 5**). We have also included citations for the studies that provided the *M. tuberculosis* genomes utilized by our study. Many of the genomes in our study come from the TB-ARC project and variants (i.e., MALI, MALI.1, TAIWAN, etc...) -- most which were uploaded to the PATRIC database in 2013. Since we were unable to find publications for some of these projects, we followed the language and reference style used in *Manson et al. 2017*⁹ when using unpublished datasets. We have now provided citations for the published studies in the first paragraph of the results section, shown below.

These strains come from a wide range of studies^{10 11 12 13 14 15 16 17 18 19 20 21 9 22 23 24 25}.

We have also provided further information in the Methods section, titled “*M. tuberculosis* strain dataset”.

References for the published and unpublished data sets can be found in **Supplementary Table 5**. The sequencing data for the TB Antibiotic Resistance Catalog (TB-ARC) projects (**Supplementary Table 5**) were generated at the Broad institute. Additional information for each of these unpublished projects can be found at the Broad Institute website (https://olive.broadinstitute.org/projects/tb_arc).

Line 554. In addition to the use of CD-hit package, the authors may want to use another established bioinformatics tool to characterise bacterial pan-genomes: (Page et al. 2015). If both CD-hit and Roary yield similar results, the readers will be more confident about the pan-genome analysis.

The CD-hit package is a tool used to cluster large amounts of protein sequences into separate groups. In fact, Roary uses CD-hit as its primary clustering tool²⁶. Following new quality control measures we instituted, recommended by reviewer 1, including removing PE/PPE genes and genes of unrealistic length (see response to Reviewer 1, comment 4, above) we find that the calculated TB pan-genome is closed and would likely be similar to that constructed by Roary. Furthermore, now that the pan-genome analysis is no longer a focus of the main text, we believe that the lack of comparisons amongst pan-genome tools will not detract from our study.

Line 571. The phylogenetic approach used by the authors is not valid. Given that the whole-genome sequences of the 1,603 strains are available, the authors must make use of SNPs at the core genome to create a whole-genome phylogeny that is comparable to that already published ((Coll et al. 2014), (Comas et al. 2013)). Selection of just a few housekeeping genes is not enough to obtain a high-resolution phylogeny required for this type of genomic studies. Furthermore, four of the seven housekeeping genes chosen (*rpoB*, *katG*, *gyrA* and *rpoC*) are likely to contain homoplastic SNPs among resistant strains that will inevitably distort the phylogeny.

We agree with the reviewer in that our phylogenetic MLST approach lacks a high-resolution necessary for rigorous downstream analysis. While our computational analysis is agnostic to phylogenetics and the use of a reference strain, we agree that our tree construction should adhere to the standards set by the community. Therefore, we have now created a phylogenetic tree of the 1595 strains using an existing and recent SNP typing scheme². Specifically, we used a total of 141 SNPs for identifying lineages and sublineages for our 1595 TB strains. These SNPs were previously determined to be sufficient for categorizing lineages². Of these SNPs, 61 were in non-synonymous sites and the other 70 were SNPs found in drug resistance genes. These 141 SNPs comprised a total of 74 genes. We used needle²⁷ to align sequences within a pan-genome cluster (a cluster is representative of a particular loci) to the H37Rv reference allele. The presence of SNPs were then used to categorize the strains into the defined lineages². Of the 1595 strains, 1366 strains were categorized and 229 were uncategorized. To categorize these genes, we built a binary SNP matrix using all of the SNPs identified from the 74 genes (885 SNPs in total), and then estimated a maximum-likelihood phylogeny using RaXML version 8²⁸, visualized below using iTOL²⁹.

Supplementary Figure 1b has now been updated with the new phylogenetic tree (shown below). Note that the tree distances are hidden in order to allow for a clear visualization of AMR phenotypes.

Line 589. The authors state that “*M. tuberculosis* has an open pan-genome.” The authors also note that “a significant portion of the unique and accessory genome was attributed to Pro-Glu (PE)-related proteins and hypothetical proteins (Fig. 1b).” These results could be the result of assembly and/or gene annotation artefacts. PE/PPE genes have a high GC content, which results in lower depth of coverage in these regions, and some members of this family of genes are highly repetitive, which make them inherently difficult to assemble from short sequence reads generated by Illumina sequencers. Fragmented assemblies at the boundaries of PE/PPE genes might lead to a higher number of predicted PE/PPE clusters. In order to rule out the possibility of assembly artefacts confounding the pan-genome analysis, I would recommend the authors repeat the pan-genome analysis using the subset of ‘Complete’ Mtb genomes sequenced using long-read sequencing (PacBio) and compare their results with those of the whole dataset. The authors should also measure the length of newly identified genes as they add more strains. Are newly identified genes shorter than core-genome genes? If so, they could be the result of miss-assemblies. The other necessary check is to quantify the amount of new DNA sequences (in base pairs) as more strains are added to make sure that the increasing number of new genes observed matches with an expected increase in new DNA sequences. The authors should perform these extra analyses before stating that “*M. tuberculosis* has an open pan-genome.” as it is established in the field that Mtb has a closed pan-genome due to the lack of inter-strain recombination.

We thank the reviewer for pointing these issues out. Based on this suggestion we have re-calculated the pan-genome curve after double checking for gene annotation artefacts. Please see our response to reviewer 1, point 4 (above) but in short we found that PE/PPE genes were significantly longer (>1 standard deviation) than the mean gene length of 1000 bps. In total, this led to the removal of 1,335 genes clusters from the pan-genome and led to a “closed” pan-genome for *M. tuberculosis*, in agreement with the literature.

Coll F, McNerney R, Guerra-Assunção JA, Glynn JR, Perdigão J, Viveiros M, Portugal I, Pain A, Martin N, Clark TG. 2014. A robust SNP barcode for typing Mycobacterium tuberculosis complex strains. *Nature Communications* 5: 4812.

Comas I, Coscolla M, Luo T, Borrell S, Holt KE, Kato-Maeda M, Parkhill J, Malla B, Berg S, Thwaites G, et al. 2013. Out-of-Africa migration and Neolithic coexpansion of Mycobacterium tuberculosis with modern humans. *Nature Genetics* 45: 1176–1182.

Page AJ, Cummins CA, Hunt M, Wong VK, Reuter S, Holden MTG, Fookes M, Falush D, Keane JA, Parkhill J. 2015. Roary: rapid large-scale prokaryote pan genome analysis. *Bioinformatics* 31: 3691–3693.

Reviewer #3 (Remarks to the Author):

The work targets an interesting and hot topic that is the investigation of AMR characteristics and evolution in MTB. To come to the conclusions the authors use statistical methods as well as machine learning approaches. Doing so the authors identified known resistance markers as well as 24 new genes somehow associated with drug resistance. A really interesting part is the analysis of interactions between genes, which is a unique feature of this study. Additionally, the authors prevent themselves of statistical artifacts by investigating the crystal structure of the newly identified genes and how those genes could play a role in AMR. The paper is well written and the results are clearly presented. The study has four main conclusions and thus the article has some length in reading. There is so many really interesting stuff

included and each step make sense, but would like a shorter version focusing on the main points of the study. The sample size and selection of data is reasonable and leads to a representative dataset. I wonder about the term “platform”, is the code somewhere provided or is there a platform where you can upload your own data and get a classification?

We thank the reviewer for closely reviewing the manuscript and appreciate their interest in our analysis of genetic interactions. Also, yes we feel this can be described as a “platform”. We have uploaded our code to a publicly accessible github repository along with a description of the workflow that should enable reproducible applications to similar GWAS-like microbial datasets. We have added the following sentence under the Methods section titled “Code Availability”,

The computational platform is provided as a github code repository (https://github.com/erolkavvas/microbial_AMR_ML/).

I would suggest to make the MLST scheme with more than just 7 housekeeping genes.

We have now constructed a phylogenetic tree of the 1595 strains using an existing and recent SNP typing scheme². The tree was inferred using 885 SNPs comprising a total of 74 genes. Supplementary Figure 1 has now been updated with the new phylogenetic tree.

In Table 1 it is a profit of the SVM compared to the ANOVA and other methods, but to be honest when *inhA* would not be associated with INH resistance the method needs definitely a reconstruction.

We were also surprised that *inhA* was not found to be significantly associated with INH in the pairwise ANOVA test. The reason for this absence becomes much more apparent when viewing the actual frequencies of resistant-dominant *inhA* alleles in Figure 4, which turn out to be very minor. Thus, perhaps it's not a surprise that GWAS approaches do not pick up *inhA* as a signal, including those that explicitly account for geographic structure (Earle et al. 2016). Importantly, the ensemble SVM approach successfully picks up *inhA* as the second highest signal, exemplifying the ability to uncover hidden signals.

The key points of the study needs to be more precisely described e.g. focusing on only 1-2 major points and skipping for example the county distribution.

We have moved both Figure 1 and the pan-genome analysis to the supplementary text as recommended by the other reviewers (Figure 1 is now Supplementary Figure 2, and the pan-genome analysis is located under Supplementary Note). We believe that this makes the paper much more focused and clear.

References

1. Tishby, N., Pereira, F. C. & Bialek, W. The information bottleneck method. *arXiv [physics.data-an]* (2000).
2. Coll, F. *et al.* A robust SNP barcode for typing Mycobacterium tuberculosis complex strains. *Nat. Commun.* **5**, 4812 (2014).
3. Farhat, M. R. *et al.* Genomic analysis identifies targets of convergent positive selection in drug-resistant Mycobacterium tuberculosis. *Nat. Genet.* **45**, 1183–1189 (2013).
4. Safi, H. *et al.* Evolution of high-level ethambutol-resistant tuberculosis through interacting mutations in decaprenylphosphoryl-[beta]-D-arabinose biosynthetic and utilization pathway genes. *Nat. Genet.* **45**, 1190–1197 (2013).
5. Earle, S. G. *et al.* Identifying lineage effects when controlling for population structure improves power in bacterial association studies. *Nat Microbiol* **1**, 16041 (2016).
6. Lees, J. A. *et al.* Sequence element enrichment analysis to determine the genetic basis of bacterial phenotypes. *Nat. Commun.* **7**, 12797 (2016).
7. Jaillard, M. *et al.* Representing Genetic Determinants in Bacterial GWAS with Compacted De Bruijn Graphs. *bioRxiv* 113563 (2017). doi:10.1101/113563
8. Kapopoulou, A., Lew, J. M. & Cole, S. T. The MycoBrowser portal: a comprehensive and manually annotated resource for mycobacterial genomes. *Tuberculosis* **91**, 8–13 (2011).
9. Manson, A. L. *et al.* Genomic analysis of globally diverse Mycobacterium tuberculosis strains provides insights into the emergence and spread of multidrug resistance. *Nat. Genet.* **49**, 395–402 (2017).
10. Miyoshi-Akiyama, T., Matsumura, K., Iwai, H., Funatogawa, K. & Kirikae, T. Complete annotated genome sequence of Mycobacterium tuberculosis Erdman. *J. Bacteriol.* **194**, 2770 (2012).
11. Roetzer, A. *et al.* Whole genome sequencing versus traditional genotyping for investigation of a Mycobacterium tuberculosis outbreak: a longitudinal molecular epidemiological study. *PLoS Med.* **10**, e1001387 (2013).
12. Wu, W. *et al.* A genome-wide analysis of multidrug-resistant and extensively drug-resistant strains of Mycobacterium tuberculosis Beijing genotype. *Mol. Genet. Genomics* **288**, 425–436 (2013).
13. Majid, M. *et al.* Genomes of Two Clinical Isolates of Mycobacterium tuberculosis from Odisha, India.

- Genome Announc.* **2**, (2014).
14. Ng, K. P. *et al.* Draft Genome Sequence of the First Isolate of Extensively Drug-Resistant (XDR) *Mycobacterium tuberculosis* in Malaysia. *Genome Announc.* **1**, (2013).
 15. Lin, N., Liu, Z., Zhou, J., Wang, S. & Fleming, J. Draft genome sequences of two super-extensively drug-resistant isolates of *Mycobacterium tuberculosis* from China. *FEMS Microbiol. Lett.* **347**, 93–96 (2013).
 16. Lanzas, F., Karakousis, P. C., Sacchetti, J. C. & Ioerger, T. R. Multidrug-resistant tuberculosis in panama is driven by clonal expansion of a multidrug-resistant *Mycobacterium tuberculosis* strain related to the KZN extensively drug-resistant *M. tuberculosis* strain from South Africa. *J. Clin. Microbiol.* **51**, 3277–3285 (2013).
 17. Cohen, K. A. *et al.* Evolution of Extensively Drug-Resistant Tuberculosis over Four Decades: Whole Genome Sequencing and Dating Analysis of *Mycobacterium tuberculosis* Isolates from KwaZulu-Natal. *PLoS Med.* **12**, e1001880 (2015).
 18. Ismail, A. *et al.* Draft Genome Sequence of a Clinical Isolate of *Mycobacterium tuberculosis* Strain PR05. *Genome Announc.* **1**, (2013).
 19. Karuthedath Vellarikkal, S. *et al.* Draft Genome Sequence of a Clinical Isolate of Multidrug-Resistant *Mycobacterium tuberculosis* East African Indian Strain OSDD271. *Genome Announc.* **1**, (2013).
 20. Al Rashdi, A. S. A., Jadhav, B. L., Deshpande, T. & Deshpande, U. Whole-Genome Sequencing and Annotation of a Clinical Isolate of *Mycobacterium tuberculosis* from Mumbai, India. *Genome Announc.* **2**, (2014).
 21. Winglee, K. *et al.* Whole Genome Sequencing of *Mycobacterium africanum* Strains from Mali Provides Insights into the Mechanisms of Geographic Restriction. *PLoS Negl. Trop. Dis.* **10**, e0004332 (2016).
 22. Merker, M. *et al.* Evolutionary history and global spread of the *Mycobacterium tuberculosis* Beijing lineage. *Nat. Genet.* **47**, 242–249 (2015).
 23. Isaza, J. P. *et al.* Whole genome shotgun sequencing of one Colombian clinical isolate of *Mycobacterium tuberculosis* reveals DosR regulon gene deletions. *FEMS Microbiol. Lett.* **330**, 113–120 (2012).

24. Cole, S. T. *et al.* Deciphering the biology of *Mycobacterium tuberculosis* from the complete genome sequence. *Nature* **393**, 537–544 (1998).
25. Camus, J.-C., Pryor, M. J., Médigue, C. & Cole, S. T. Re-annotation of the genome sequence of *Mycobacterium tuberculosis* H37Rv. *Microbiology* **148**, 2967–2973 (2002).
26. Page, A. J. *et al.* Roary: rapid large-scale prokaryote pan genome analysis. *Bioinformatics* **31**, 3691–3693 (2015).
27. Rice, P., Longden, I. & Bleasby, A. EMBOSS: the European Molecular Biology Open Software Suite. *Trends Genet.* **16**, 276–277 (2000).
28. Stamatakis, A. RAxML version 8: a tool for phylogenetic analysis and post-analysis of large phylogenies. *Bioinformatics* **30**, 1312–1313 (2014).
29. Letunic, I. & Bork, P. Interactive tree of life (iTOL) v3: an online tool for the display and annotation of phylogenetic and other trees. *Nucleic Acids Res.* **44**, W242–5 (2016).

Reviewers' comments:

Reviewer #2 (Remarks to the Author):

I am happy to see that the authors have correctly addressed most of my comments. There are still two outstanding issues that the authors should address:

1. The authors should make use genome-wide SNPs at the core genome, not only lineage/sub-lineage defining SNPs, to create a robust phylogeny. It is accepted to estimate a maximum-likelihood phylogeny using RAxML and visualize it using iTOL. Please do not hide the branch lengths when visualizing the tree.

2. I am happy to see that katG allele 3 and oxcA allele 2 co-occur in different lineages/genetic backgrounds, which rules out they are the result of linkage. This was just an example, the authors should do the same analysis for all co-occurring alleles and find a way to summarise these results in Figure 2 (previous Figure 3) and Supplementary Tables alike. The authors could add in how many lineages each pair of alleles co-occur, maybe in brackets in each cell in Figure 2, or alternatively add a superscript to indicate whether each pair of alleles co-occur in the same lineage or across multiple lineages. The latter will provide further evidence of a putative epistatic interaction.

comments on the authors' answers to Reviewer 1 comments

Major issue 1 - Reviewer 1 highlights that fact that Mycobacterium tuberculosis strains are rarely exposed to just one drug, which results in drug resistances tend to co-occur in the same strains. This makes it hard to tell which gene associates with which drug. Although this is not a limitation of this study per se, the authors do not address this issue and acknowledge that "it would be interesting and valuable to deconvolve out an association of a region with a drug. Our study, however, does not provide a solution to this problem." The authors should include this text in the Discussion section of the main manuscript to highlight this limitation.

Major issue 2 - Reviewer 1 highlights the same issue I highlighted in my revision, that is, the authors need to make sure that epistatic alleles are homoplastic (they occur across different lineage/sub-lineage) to provide further evidence of their putative epistatic interaction. My comment on this matter was:

"I am happy to see that katG allele 3 and oxcA allele 2 co-occur in different lineages/genetic backgrounds, which rules out they are the result of linkage. This was just an example, the authors should do the same analysis for all co-occurring alleles and find a way to summarise these results in Figure 2 (previous Figure 3) and Supplementary Tables alike. The authors could add in how many lineages each pair of alleles co-occur, maybe in brackets in each cell in Figure 2, or alternatively add a superscript to indicate whether each pair of alleles co-occur in the same lineage or across multiple lineages. The latter will provide further evidence of a putative epistatic interaction."

Major issue 3 - Both reviewer 1 and the authors themselves highlight another limitation of the study, that is: "The structural mutation mapping does indeed lack both quantitative rigor and a biological basis in providing confidence to the selected AMR alleles" and that "We do not claim that it added any more support to the causality of an implicated allele". The authors should include this text in the Discussion section of the main manuscript to highlight this limitation.

Major issue 4 - Correctly addressed.

Other issues 1 - Correctly addressed.

Other issues 2 - Correctly addressed.

Reviewer #3 (Remarks to the Author):

The manuscript reads well. The authors incorporated all suggestions made. I have no further suggestions

This font color represents our comment to reviewers

This highlight represents the text that has been changed in the main manuscript or supplementary text.

Reviewers' comments:

Reviewer #1 (Remarks to the Author):

Reviewers' comments:

Reviewer #2 (Remarks to the Author):

I am happy to see that the authors have correctly addressed most of my comments. There are still two outstanding issues that the authors should address:

We thank reviewer #2 for the very constructive feedback. Addressing these comments has greatly improved this paper.

1. The authors should make use genome-wide SNPs at the core genome, not only lineage/sub-lineage defining SNPs, to create a robust phylogeny. It is accepted to estimate a maximum-likelihood phylogeny using RAxML and visualize it using iTOL. Please do not hide the branch lengths when visualizing the tree.

We have now extended our analysis of phylogeny beyond the SNP barcodes to the genome-wide SNPs present in the core genome. Specifically, we expanded our SNP list from the 71 SNP-barcode genes (Coll et al. 2016) to a set of 2803 core genes that appeared in at least 1593 strains including the H37Rv reference strain (83332.12). These 2803 genes approximate the core genome and are comprised of a total of 21,206 SNPs. We then constructed a phylogeny with these SNPs using RAxML using the GTRGAMMA parameter and visualized the tree using iTOLv3. As recommended, the branch lengths are no longer hidden in the tree visualization. The robust phylogenetic tree is portrayed below along with the newly edited version of **Supplementary Figure 1** that includes the new phylogenetic tree. We have included the phylogenetic tree as supplementary material (**Supplementary File 5**).

2. I am happy to see that *katG* allele 3 and *oxcA* allele 2 co-occur in different lineages/genetic backgrounds, which rules out they are the result of linkage. This was just an example, the authors should do the same analysis for all co-occurring alleles and find a way to summarise these results in Figure 2 (previous Figure 3) and Supplementary Tables alike. The authors could add in how many lineages each pair of alleles co-occur, maybe in brackets in each cell in Figure 2, or alternatively add a superscript to indicate whether each pair of alleles co-occur in the same lineage or across multiple lineages. The latter will provide further evidence of a putative epistatic interaction.

We have now provided a supplementary excel sheet describing the distribution of lineages for each allele-allele pair across all gene-gene pairs identified in our epistasis analysis (**Supplementary Table 6**). In addition, we followed the visual recommendation recommended by the reviewer and added a numeric subscript describing the number of unique sublineages for each allele-allele pair. We determined the number of unique sub-lineages to be the maximum number of sub-lineages that occurs at a single lineage/sublineage branch point, amongst all potential branch points in the set of lineage/sublineages

captured by the allele co-occurrence. The co-occurrence tables in **Supplementary Data File 4** now reflect this change. **Figure 2** in the main text (shown below) now includes the subscripts.

We additionally provided a note of the additional lineage material in the methods section, titled “Phylogenetic Tree and categorization of lineages”, with the following text:

“The frequency of lineage variants are displayed as subscripts to help discern between epistatic alleles and those in tight linkage (**Supplementary Table 6**). Implicated co-occurring alleles that span different lineages are unlikely to be in tight linkage (i.e., hitchhikers). We determined the lineages of our set of *M. tuberculosis* strains using previously defined lineage/sub-lineage SNPs (Coll et al. 2014).”

Comments on the authors’ answers to Reviewer 1 comments

Major issue 1 - Reviewer 1 highlights that fact that Mycobacterium tuberculosis strains are rarely exposed to just one drug, which results in drug resistances tend to co-occur in the same strains. This makes it hard to tell which gene associates with which drug. Although this is not a limitation of this study per se, the authors do not address this issue and acknowledge that “it would be interesting and valuable to deconvolve out an association of a region with a drug. Our study, however, does not provide a solution to this problem.” The authors should include this text in the Discussion section of the main manuscript to highlight this limitation.

We have added text describing this limitation in our discussion section,

"While our framework successfully identifies genetic AMR signatures, there are limitations to the approach that future efforts may expand upon. For one, our platform utilizes prior knowledge of known gene-antibiotic relationships limiting its ability to uniquely deconvolve regional associations with a specific drug (**Supplementary Note**)."

Major issue 2 - Reviewer 1 highlights the same issue I highlighted in my revision, that is, the authors need to make sure that epistatic alleles are homoplastic (they occur across different lineage/sub-lineage) to provide further evidence of their putative epistatic interaction. My comment on this matter was:

"I am happy to see that katG allele 3 and oxcA allele 2 co-occur in different lineages/genetic backgrounds, which rules out they are the result of linkage. This was just an example, the authors should do the same analysis for all co-occurring alleles and find a way to summarise these results in Figure 2 (previous Figure 3) and Supplementary Tables alike. The authors could add in how many lineages each pair of alleles co-occur, maybe in brackets in each cell in Figure 2, or alternatively add a superscript to indicate whether each pair of alleles co-occur in the same lineage or across multiple lineages. The latter will provide further evidence of a putative epistatic interaction."

See above.

Major issue 3 - Both reviewer 1 and the authors themselves highlight another limitation of the study, that is: "The structural mutation mapping does indeed lack both quantitative rigor and a biological basis in providing confidence to the selected AMR alleles" and that "We do not claim that it added any more support to the causality of an implicated allele". The authors should include this text in the Discussion section of the main manuscript to highlight this limitation.

We have added text describing this limitation in our discussion section,

"... regional associations with a specific drug (**Supplementary Note**). In addition, while our structural analysis provides a foundation for hypothesizing evolutionary drivers, it does not prove causality of an allele. These results should be interpreted as the delineation of susceptible and resistant alleles into distinct structural features that can be leveraged in the future for experimental validation."

These

Major issue 4 - Correctly addressed.

Other issues 1 - Correctly addressed.

Other issues 2 - Correctly addressed.

Reviewer #3 (Remarks to the Author):

The manuscript reads well. The authors incorporated all suggestions made. I have no further suggestions

REVIEWERS' COMMENTS:

Reviewer #2 (Remarks to the Author):

I am happy with the latest changes incorporated by the authors. The few remaining concerns were correctly addressed. I have no further suggestions.